# Meent: Differentiable Electromagnetic Simulator for Machine Learning

## Abstract

Electromagnetic (EM) simulation plays a crucial role in analyzing and designing devices with sub-wavelength scale structures such as semiconductor devices and future displays. Specifically, optics problems such as estimating semiconductor device structures and designing nanophotonic devices provide intriguing research topics with far-reaching real world impact. Traditional algorithms for such tasks require iteratively refining parameters through simulations, which often yield sub-optimal results due to the high computational cost of both the algorithms and EM simulations. Machine learning (ML) emerged as a promising candidate to mitigate these challenges, and optics research community has increasingly adopted ML algorithms to obtain results surpassing classical methods across various tasks. To foster a synergistic collaboration between the optics and ML communities, it is essential to have an EM simulation software that is user-friendly for both research communities. To this end, we present `meent`, an EM simulation software that employs rigorous coupled-wave analysis (RCWA). Developed in Python and equipped with automatic differentiation (AD) capabilities, `meent` serves as a versatile platform for integrating ML into optics research and vice versa. To demonstrate its utility as a research platform, we present three applications of `meent`: 1) generating a dataset for training neural operator, 2) serving as an environment for the reinforcement learning of nanophotonic device optimization, and 3) providing a solution for inverse problems with gradient-based optimizers. These applications highlight `meent`'s potential to advance both EM simulation and ML methodologies. The code is available on our Github repository with the MIT license to promote the cross-polinations of ideas among academic researchers and industry practitioners.

## 1 Introduction

Harnessing light-matter interaction to design or analyze a device with sub-wavelength scale structure has a wide range of applications, including high-efficiency solar cells (Peter Amalathas & Alkaisi, 2019; Snaith, 2013), ultra-thin metalenses and displays (Zhang et al., 2018; Aieta et al., 2012), optical metrology for semiconductor fabrication (Timoney et al., 2020; Den Boef, 2013), X-ray diffraction for material analysis (von Laue, 1915; Norton et al., 1998), optical computation (Fang et al., 2005; Silva et al., 2014), and so on. Their implication to the real-world is far-reaching, leading to improved renewable energy production, enhanced user experience, and next-generation computation. Electromagnetic (EM) simulation plays a crucial role in such applications, which also poses a challenging problem due to its time-consuming nature for precise calculation (Burger et al., 2008) and iterative characteristic of optimization.

Machine learning (ML) is a promising candidate to solve such a problem, and with the advent of deep learning, optics community has been made successful efforts (Malkiel et al., 2018; Melati et al., 2019; Jiang & Fan, 2019) to leverage modern machine learning techniques to find both better optimization algorithms (So & Rho, 2019; Seo et al., 2021; Jin et al., 2020; Colburn & Majumdar, 2021; Kim & Lee, 2023) and faster electromagnetic simulators (Raissi et al., 2019; Li et al., 2020a; Lu et al., 2021). These researches show significant potential of ML for uncovering new insights and expediting scientific discoveries. On the other hand, from the viewpoint of ML, designing or analyzing such devices are ideal environments for developing new ML algorithms, as they provide

Figure 1: **Summary.** *Simulation Algorithm* depicts the process flow of electromagnetic simulation algorithm, namely RCWA, in `meent`. The physical environment to be simulated is described by formulas about light and matter, and sent to Fourier space. Using convolution operation, the electric and magnetic fields are described with the transformation matrices, $\Omega_L$ and $\Omega_R$. These two equations are combined into one, and a general solution is found by eigendecomposition. Applying the boundary conditions, the particular solution can be found. Using this solution, we can find the transmittance, reflectance, and field distribution. *Applications* presents the most representative domains that `meent` can be utilized. Beam deflector and color router are metasurface-related applications that control the direction of propagating light. Optical metrology is used to estimate the property of the specimen. It can be a structure of a device, material property of a material, etc.

both ample simulation data to train ML models and well-defined optimization goals, often called figure of merit (FoM), for the real-world applications.

However, the integration of ML into computational optics presents several challenges. Traditional EM simulation software are often written in languages like C or MATLAB and therefore cannot be seamlessly integrated into ML packages mostly developed for Python ecosystem. This results in the loss of the computational graph needed for automatic differentiation (AD), which is useful for gradient-based optimization. Furthermore, the scarcity of public data in certain domains, such as semiconductor fabrication industry, compounded by stringent intellectual property regulations, poses significant obstacles, especially to ML researchers lacking domain expertise. Overcoming these barriers requires innovative approaches in generating and sharing data to enable ML researchers to explore new frontiers in computational optics.

In response to these challenges, we introduce `meent`, a Python-native differentiable EM simulator. `meent` is based on rigorous coupled-wave analysis (RCWA) (Moharam & Gaylord, 1981; Moharam et al., 1995a;b; Lalanne & Morris, 1996; Granet & Guizal, 1996; Li, 1996; Rumpf, 2006; Li, 2014), a high-throughput, deterministic EM simulation algorithm that is widely adopted in optics across academia and industries. Key features of `meent` include its compatibility with automatic differentiation (AD) (Baydin et al., 2018; Moses & Churavy, 2020) for modeling and optimizing devices in a continuous space. AD compatibility in ML toolchain is pivotal, and while some existing tools support vector modeling and others support AD, none offer both functionalities simultaneously. For developer ergonomics, `meent` is developed to be compatible with three different backend frameworks: NumPy (Harris et al., 2020), JAX (Bradbury et al., 2018), and PyTorch (Paszke et al., 2019). By supporting multiple backends, `meent` facilitates easy adoption among researchers with varying levels of domain expertise and different backend preferences.

We showcase the utility of `meent` with various applications of ML to optics. First we present how to use `meent` to analyze and design a metasurface, whose sub-wavelength scale structure is carefully designed to achieve unprecedented control of light (Kildishev et al., 2013; Yu & Capasso, 2014; Sun et al., 2019). We also illustrate how to use `meent` in optical metrology (Zuo et al., 2022), one of the most successful industrial applications within the semiconductor fabrication, that serves to estimate the dimensions of device structures between process steps, thereby effectively monitoring excursions and maximizing yield due to its non-destructive nature and high throughput capabilities.

By enabling each user to generate datasets tailored to specific research needs, `meent` can democratize the access to EM simulation data. We hope that `meent` will facilitate collaboration between ML and optics researchers and thereby accelerate scientific discovery in computational optics.

Our contributions are summarized as follows:

- Development of `meent`, a Python-native EM simulation software under MIT license supporting automatic differentiation and continuous space operation in ML frameworks.
- Demonstration of `meent`'s versatility with examples of ML algorithms, including Fourier neural operator, model-based RL, and gradient-based optimizers.
- Documentation of `meent` with detailed explanations and instructions.

## 2 RELATED WORK

**EM simulation algorithms.** There exist several methods for full-wave[1] EM simulation, each offering distinct advantages. Here, we review finite difference time domain (FDTD) and rigorous coupled wave analysis (RCWA). FDTD operates within the real space and time domain, employing the finite difference method. It discretizes space into grids and iteratively solves the function at these grid points over successive time steps (Taflove, 1980). RCWA operates in reciprocal space and frequency domain, which requires two conditions for Fourier analysis: time-harmonic field[2] and periodicity of a structure.

Table 1: **FDTD and RCWA**

|  | space | domain | type | throughput |
|---|---|---|---|---|
| FDTD | real space | time | numerical | low |
| RCWA | reciprocal space | frequency | semi-analytical | high |

Table 1 shows a comparative analysis of FDTD and RCWA. Both methods solve Maxwell's equations, but they operate in different domains, as explained. FDTD is a fully numerical method, whereas RCWA is considered semi-analytic, as it allows for analytical solutions of the fields in the propagation direction. FDTD is general but RCWA is applicable to specific cases where the fields are time-harmonic and the structure has periodicity. By losing the generality, RCWA can show much faster simulations compared to FDTD for many practical cases.

Notable open-source software packages for FDTD include Meep (Oskooi et al., 2010), gprMAX (Warren et al., 2016), OpenEMS (Liebig), ceviche (Hughes et al., 2019) and FDTD++. In the realm of RCWA simulators, Reticolo (Hugonin & Lalanne, 2021) and S4 (Liu & Fan, 2012), implemented in MATLAB and C++ respectively, have earned recognition and been extensively utilized in numerous research endeavors. With the emergence of ML, the significance of Python-native code has grown substantially, prompting optics researchers to familiarize themselves with Python and its associated technologies. gRCWA (Jin et al., 2020), rcwa-tf (Colburn & Majumdar, 2021), and TORCWA (Kim & Lee, 2023) are notable for their support of AD. Comparing `meent` to these AD-enabled tools, the main novelty is the vector-type modeling which enables modeling in continuous space while the others reside in discrete space which critically limits the resolution of modeling hence of optimization algorithm. Additionally, the inverse rule for Fourier analysis (Li, 1996; 2014) is applied to improve the convergence of TM polarized light. Table 2 summarizes supporting features of each EM simulation solver that provides automatic differentiation.

---

[1]Full-wave simulations solve the exact Maxwell's equations without relying upon simplifying assumptions.
[2]The electric and magnetic fields at any location vary sinusoidally with time.

Table 2: **Automatic differentiation enabled solvers and their features**

|  | raster input | vector input | GPU | inverse rule | backend |
|---|---|---|---|---|---|
| grcwa | O | X | X | X | NumPy |
| rcwa-tf | O | X | O | X | TensorFlow |
| torcwa | O | X | O | X | PyTorch |
| meent | O | O | O | O | NumPy, JAX, PyTorch |

**ML applications in optics.** Assisted with physical simulators, ML is being actively embraced across scientific domains to substitute heavy simulations with deep models that serve as surrogate solvers, offering high throughput and increased robustness to hidden noise. Seminal works such as physics-informed neural network (PINN) (Raissi et al., 2017a;b; 2019) and neural operators (Li et al., 2020a; Cai et al., 2021; Li et al., 2020b; 2023; Lu et al., 2021; 2022; Jin et al., 2022) showed their potential as surrogate EM solvers (Pestourie et al., 2020; Kim et al., 2021). Reinforcement learning (RL) also showed its efficacy in the scientific domains, such as magnetic control of tokamak plasmas (Degrave et al., 2022) and classical mechanics (Lillicrap et al., 2015; Todorov et al., 2012).

Representative examples of surrogate EM solver include MaxwellNet (Lim & Psaltis, 2022), an instance of PINN. Fourier neural operator was used in (Augenstein et al., 2023), where optimization of nanophotonic device (Park et al., 2022) is also conducted. Deep generative model was used in (So & Rho, 2019) to reduce computational cost compared to traditional optimization algorithm, and the feasibility of using model-free RL was demonstrated in (Sajedian et al., 2019; Seo et al., 2021; Park et al., 2024). Our example explores the possibility of applying model-based RL to device optimization, rooted on RNN-based world model (Ha & Schmidhuber, 2018; Hafner et al., 2019b;a; 2020; 2023).

**Datasets and benchmarks in nanophotonics.** Efforts have been made to create and release datasets to engage machine learning researchers in nanophotonics. (Kim et al., 2023; Yang et al., 2023) offer datasets generated from EM simulators and evaluate ML techniques for inverse problems in metasurface design. Our work not only offers a set of codes for benchmarking ML algorithms but also includes the solver itself, which is essential for a complete and comprehensive simulation cycle.

## 3 MEENT: ELECTROMAGNETIC SIMULATION FRAMEWORK

**Electromagnetic simulation algorithm.** meent uses rigorous coupled wave analysis (RCWA) (Moharam & Gaylord, 1981; Moharam et al., 1995a;b; Lalanne & Morris, 1996; Granet & Guizal, 1996; Li, 1996; Rumpf, 2006; Li, 2014), which is based on Faraday's law and Ampére's law of Maxwell's equations (Kim et al., 2012),

$$\nabla \times \mathbf{E} = -j\omega\mu_0\mathbf{H}, \qquad \nabla \times \mathbf{H} = j\omega\varepsilon_0\varepsilon_r\mathbf{E}, \tag{1}$$

where $\mathbf{E}$ and $\mathbf{H}$ are electric and magnetic field in real space, $j$ denotes the imaginary unit number, i.e. $j^2 = -1$, $\omega$ denotes the angular frequency, $\mu_0$ is vacuum permeability, $\varepsilon_0$ is vacuum permittivity, and $\varepsilon_r$ is relative permittivity. As illustrated in Figure 1, it is a technique used to solve PDEs in *Fourier space*, aiming to estimate optical properties such as diffraction efficiency or field distribution. We reserve the detail of RCWA for Appendices A and G for readers interested in delving into the fundamentals of RCWA.

**Geometry modeling.** meent offers support for two modeling types: raster and vector. Analogous to the image file format, raster-type represents data as an array, while vector-type utilizes a set of objects, with each object comprising vertices and edges, as shown in Figure 2a. Due to their distinct formats, each method employs different algorithms for space transformation, resulting in different types of geometry derivatives, including topological and shape derivatives, as depicted in Figure 2b. The topological derivative yields the gradient with respect to the permittivity changes of every cell in the grid, while the shape derivative provides the gradient with respect to the deformations of a shape.

These two modeling methods offer distinct advantages and are suited to different applications: raster modeling is ideal for freeform metasurface design, where pixel-wise operations are utilized, while

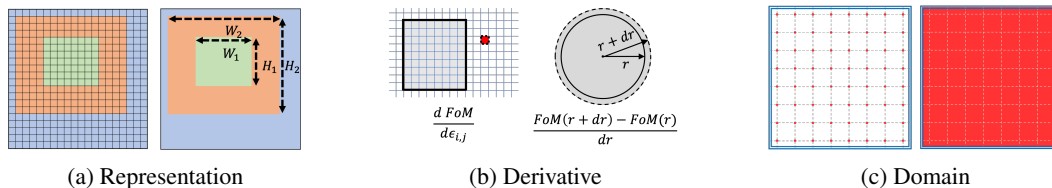

(a) Representation           (b) Derivative           (c) Domain

Figure 2: **Characteristics of each modeling type.** In each subfigure, the left side depicts raster while the right side depicts vector. (a) illustrates how the geometry is formed by each method. (b) presents a schematic diagram highlighting the difference between the topological derivative (left) and shape derivative (right). (c) The area enclosed by the blue double line denotes the codomain, while the red dots on the left and red area on right represent the range.

vector modeling is more appropriate for OCD metrology, where object dimensions are defined in continuous space, as illustrated in Figure 2c.

**Fourier analysis.** `meent` provides three methods for Fourier series: discrete Fourier series (DFS), enhanced DFS (EFS) and Fourier series on piecewise-constant function (here we call it continuous Fourier series, CFS). In DFS, the function $\varepsilon(x, y)$ to be transformed is sampled at a finite number of points, and this means it's given in matrix form with rows and columns, $\varepsilon_{\mathtt{r},\mathtt{c}}$. The coefficients from DFS are then given by this equation:

$$c_{n,m} = \frac{1}{P_x P_y} \sum_{\mathtt{c}=0}^{P_x-1} \sum_{\mathtt{r}=0}^{P_y-1} \varepsilon_{\mathtt{r},\mathtt{c}} \cdot \exp\left[-j \cdot 2\pi \left(\frac{m}{P_x}\mathtt{c} + \frac{n}{P_y}\mathtt{r}\right)\right], \tag{2}$$

where $c_{n,m}$ is the Fourier coefficients ($m^{th}$ in X and $n^{th}$ in Y), $P_x, P_y$ are the sampling frequency (the size of the array), $\varepsilon_{\mathtt{r},\mathtt{c}}$ is the $(\mathtt{r}, \mathtt{c})^{th}$ element of the permittivity array. Here, the sampling frequency ($P_x, P_y$) is very important (Smith, 1999; Antoniou, 2005; Kreyszig et al., 2011). If this is not enough, an aliasing occurs: DFS cannot correctly capture the original signal. To address this issue, `meent` offers EFS, which performs upscaling of the input data to produce simulation results that more closely align with those obtained from CFS. Continuous Fourier series utilizes the entire function to find the coefficients while DFS uses only some of them (through sampling). The Fourier coefficients can be expressed as follow:

$$c_{n,m} = \frac{1}{\Lambda_x \Lambda_y} \int_{x_0}^{x_0+\Lambda_x} \int_{y_0}^{y_0+\Lambda_y} \varepsilon(x, y) \cdot \exp\left[-j \cdot 2\pi \left(\frac{m}{\Lambda_x}x + \frac{n}{\Lambda_y}y\right)\right] dy dx, \tag{3}$$

where $\Lambda_x, \Lambda_y$ are the period of the unit cell.

**Simulation accuracy.** The simulation accuracy is compared to Reticolo in the context of designing a one-dimensional metasurface beam deflector. Reticolo is a well-established classical RCWA tool with a long history and extensive adoption within the optics community. Comprehensive details are provided in the appendix I.

Over 600,000 structures were simulated using four different RCWA implementations (CFS, DFS, and EFS in `meent` and Reticolo). Using Reticolo as a reference, the diffraction efficiency of transmission, which ranges between 0 and 1, was compared. Figure 3 presents a histogram of the discrepancies from the Reticolo results. The CFS demonstrates the smallest errors, with a median discrepancy of $2.1 \times 10^{-14}$, attributed to the fact that Reticolo also employs CFS (the CFS algorithm in `meent` are derived from Reticolo). In contrast, DFS exhibits the poorest matching performance; however, this can be improved with EFS. The median discrepancy for DFS is $4.3 \times 10^{-4}$, which decreases to $1.4 \times 10^{-7}$ when using EFS.

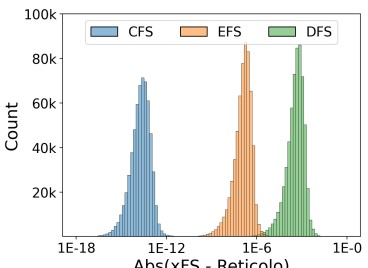

Figure 3: **Accuracy by Fourier analysis type.** Histogram of the difference compared to Reticolo.

## 4 MEENT IN ACTION: MACHINE LEARNING ALGORITHMS APPLIED TO OPTICS PROBLEMS

We have prepared six applications: the first three focus on investigating machine learning (ML) algorithms in optics problems, while the remaining three focus on the development of nanophotonic devices. The final three examples are presented in Appendix J, while the first three are discussed in this section. First, we explore neural PDE solvers for Maxwell's equations, using `meent` as a data generator. Then we delve into device optimization through reinforcement learning (RL), utilizing `meent` as an RL environment. Lastly, we address inverse problems within the semiconductor metrology domain, leveraging `meent` as a comprehensive solution for both simulation and optimization.

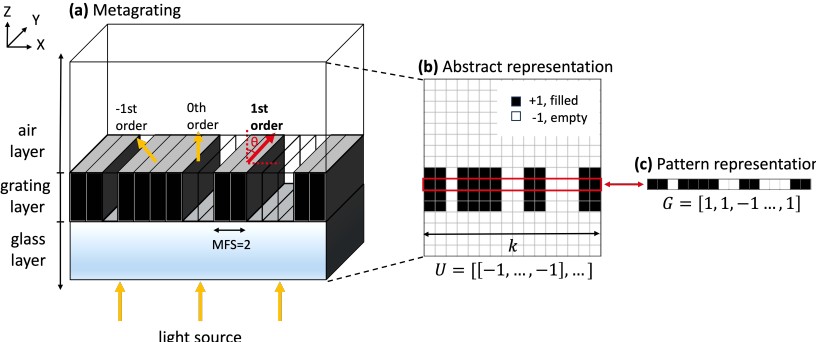

Figure 4: **Metagrating and its representations.** (a) An example of metagrating sized 16 cells, where the grating layer is bounded by air and glass layers. (b) Abstract representation of (a), called $U$. (c) Array representation of the grating pattern, expressed as $G$.

Throughout Sections 4.1 and 4.2, analysis and design of metagrating beam deflector are performed. A metagrating is a specific type of metasurface that is arranged in a periodic pattern and is primarily used to direct light into specific angle $\theta$ as shown in Figure 4. At the grating layer, a material is placed on $k$ uniform *cells*, and has the constraint of minimum feature size (MFS). MFS refers to the smallest contiguous grating cells the device can have. Our figure of merits (FoMs) from the beam deflector include deflection efficiency $\eta \in [0, 1]$ and $x$-component of electric field $\mathbf{E}$.

### 4.1 FOURIER NEURAL OPERATOR: PREDICTION OF ELECTRIC FIELD DISTRIBUTION

We provide two representative baselines of neural PDE solvers for predicting electric field: image-to-image model, UNet (Ronneberger et al., 2015) and operator learning model, Fourier neural operator (FNO) (Li et al., 2020a).

**Problem setup.** Our governing PDE that describes electric field distribution can be found by substituting left-hand side into right-hand side of Equation 1,

$$\nabla \times \nabla \times \mathbf{E} = \omega^2 \mu_0 \varepsilon_0 \varepsilon_r \mathbf{E}. \tag{4}$$

The objective of our neural PDE solver is to predict the electric field corresponding to a given grating pattern based on Maxwell's equations in the transverse magnetic (TM) polarization case. To solve this, let $\mathcal{O}$ be an operator that maps a function $u(x)$ to another function $v(x)$, such that $v(x) = \mathcal{O}(u)(x)$. We aim to find an approximator for $\mathcal{O}$, which will be represented by a neural network. In this example, we define the following notations: $g(x)$ represents a function that describes the grating pattern, while $u(x)$ denotes a function that characterizes the physical environment, which includes $g(x)$. Furthermore, $v(x)$ refers to the $x$-component of the electric field distribution associated with $u(x)$. The variables $G \in \{1, -1\}^k$, $U \in \{1, -1\}^{k \times k}$, and $V \in \mathbb{R}^{2 \times k \times k}$ represent the discretized values of $g(x)$, $u(x)$, and $v(x)$, respectively.

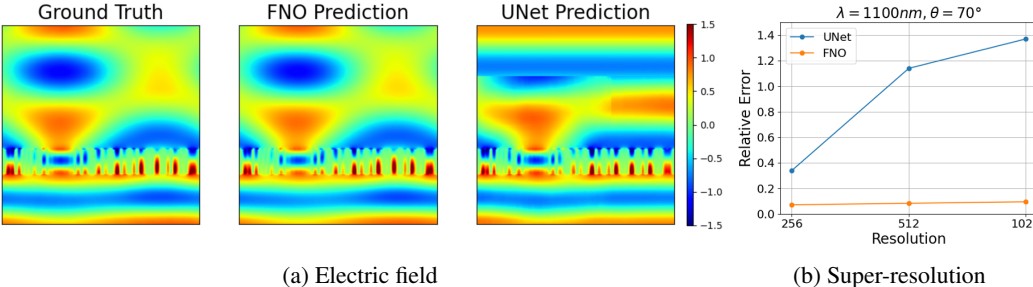

(a) Electric field             (b) Super-resolution

Figure 5: **FNO's approximation of Maxwell's equation.** (a) Real parts of electric field distribution, (left) ground truth, (middle) prediction from FNO and (right) prediction from UNet. FNO is able to predict overall field distribution and also the grating area, but UNet fails. (b) Test result on higher resolutions of fields, $512 \times 512$ and $1024 \times 1024$. The models were trained on $256 \times 256$ resolution. FNO shows little increase in the test error for predicting fields in higher resolution whereas UNet shows huge increase.

The dataset preparation begins by sampling a grating pattern $G$ from a uniform distribution, such that $G = [e_1, ..., e_k] \sim \text{Unif}(\{-1, 1\})$ while adhering to the constraint of MFS[3]. This pattern is then padded with $-1$ at the top and bottom layers to include the regions representing the incoming and outgoing electric fields, resulting in a matrix $U$, as shown in Figure 4b. The function `meent` solves Equation 4 for the given $U$ and returns the electric field $V$. Note that the first dimension of $V$ corresponds to the real and imaginary parts of the electric field, while the second and third dimensions, $k$, denote the dimensions of the matrix $U$. This data pair $(U, V)$ is derived from specific physical conditions—related to the wavelength $\lambda$ and deflection angle $\theta$. We generated 10,000 pairs as a training set, and repeated this process to create datasets each corresponding to nine different physical conditions.

**Fourier neural operator.** The effectiveness of FNO for solving Maxwell's equation in our meta-grating beam deflector is exhibited in Figure 5a. We follow techniques from (Augenstein et al., 2023), in which original FNO is adapted to light scattering problem by applying batch normalization (Ioffe & Szegedy, 2015), adding zero-padding to the input and adopting Gaussian linear error unit (GELU) activation (Hendrycks & Gimpel, 2016). We further improved FNO's parameter efficiency by applying Tucker factorization (Kossaifi et al., 2023), where a model's weight matrices are decomposed into smaller matrices for low-rank approximation. In addition to field prediction capability, we also show zero-shot super-resolution (trained in lower resolution, tested on higher resolution) capability in Figure 5b, which is claimed to be a major contribution of FNO (Li et al., 2023).

A model trained on a loss function is named as {Model}-{Loss function}, e.g., FNO-L2 is an FNO trained with L2 loss. $c_1 = 0.7$ and $c_2 = 0.3$ are set for RW L2 loss, emphasizing model to predict the grating area more accurately. A model is trained specifically to a single physical condition. Since the solution space of a PDE is highly dependent on physical conditions, we assessed the robustness of baseline models across nine conditions and collectively report in Table 4.

Table 3: **Loss functions.** $\| \cdot \|$ is the Euclidean norm, RW is shorthand for region-wise, and $grating, air, glass$ refers to the sets of indices for each region on matrix representation. All losses are relative error, i.e., normalized by the magnitude of the ground truth $y$.

| Name | Notation | Definition |
|------|----------|------------|
| L2 loss | $L_2$ | $\|y - \hat{y}\| \, / \, \|y\|$ |
| RW L2 loss | $L_{2,rw}$ | $c_1 \cdot L_{2,grating} + c_2 \cdot (L_{2,air} + L_{2,glass})$ |
| H1 loss | $H_1$ | $\{(\|y - \hat{y}\|^2 + \|y' - \hat{y}'\|^2) \, / \, (\|y\|^2 + \|y'\|^2)\}^{1/2}$ |

Table 4 summarizes the results of UNet and FNOs with various loss functions and metrics. Notably, FNO-L2 outperforms UNet-L2 by a substantial margin (8.71 compared to 34.80, resulting in a 76%

---

[3]MFS was chosen as 4 which is more granular than 8 in (Park et al., 2024).

Table 4: **Test error across three loss functions.** Smaller the better. Of the column names, top row is the name of the models and bottom row is the test metrics.

| Condition ($\lambda$ / $\theta$) | UNet-L2 | | | FNO-L2 | | | FNO-RW L2 | | | FNO-H1 | | |
|---|---|---|---|---|---|---|---|---|---|---|---|---|
| | L2 | RW L2 | H1 | L2 | RW L2 | H1 | L2 | RW L2 | H1 | L2 | RW L2 | H1 |
| $1100nm$ / $70°$ | 34.04 | 22.64 | 33.28 | 7.15 | 6.52 | 14.57 | 7.35 | 4.14 | 10.95 | 6.04 | 3.56 | 6.35 |
| $1100nm$ / $60°$ | 41.61 | 41.86 | 47.82 | 14.57 | 17.37 | 26.65 | 16.03 | 14.7 | 24.11 | 11.09 | 10.98 | 14.62 |
| $1100nm$ / $50°$ | 24.37 | 56.33 | 61.05 | 2.52 | 22.38 | 33.70 | 2.58 | 21.93 | 33.81 | 2.07 | 12.33 | 17.37 |
| $1000nm$ / $70°$ | 43.44 | 22.17 | 29.55 | 15.15 | 5.7 | 12.16 | 15.19 | 4.93 | 11.91 | 9.02 | 3.35 | 5.42 |
| $1000nm$ / $60°$ | 34.02 | 54.74 | 56.98 | 10.7 | 21.89 | 32.66 | 9.5 | 22.93 | 32.74 | 7.88 | 15.05 | 19.21 |
| $1000nm$ / $50°$ | 28.46 | 39.62 | 44.28 | 2.88 | 12.34 | 22.51 | 2.25 | 11.66 | 21.50 | 2.19 | 8.26 | 12.15 |
| $900nm$ / $70°$ | 40.78 | 27.21 | 34.25 | 15.14 | 8.37 | 15.05 | 13.63 | 6.51 | 12.67 | 10.8 | 5.03 | 7.31 |
| $900nm$ / $60°$ | 31.36 | 30.53 | 34.07 | 6.07 | 11.10 | 17.27 | 5.47 | 9.08 | 14.61 | 4.85 | 7.26 | 9.24 |
| $900nm$ / $50°$ | 35.11 | 51.64 | 51.59 | 4.23 | 22.87 | 30.79 | 3.77 | 19.89 | 27.33 | 3.29 | 14.91 | 17.75 |
| Mean | 34.80 | 38.53 | 43.65 | 8.71 | 14.28 | 22.81 | 8.42 | 12.86 | 21.07 | **6.36** | **8.97** | **12.16** |
| $\pm$Std | $\pm$5.95 | $\pm$12.81 | $\pm$10.79 | $\pm$5.95 | $\pm$6.58 | $\pm$7.93 | $\pm$5.12 | $\pm$6.92 | $\pm$8.47 | **$\pm$3.32** | **$\pm$4.31** | **$\pm$5.00** |

lower mean error) while utilizing only one-tenth of the parameters of UNet. This performance can be further enhanced by employing different loss functions: FNO-H1 demonstrates the best performance across all test metrics. The moderate performance of UNet in other PDE solvers (Hassan et al., 2024; Augenstein et al., 2023) contrasts with its poor performance in our task, which we attribute to its inability to capture complex dynamics around the grating area. More information on this experiment is provided in Appendix D.

## 4.2 MODEL-BASED REINFORCEMENT LEARNING: METASURFACE DESIGN

Here we demonstrate that `meent` can be used as an environment to train a model-based reinforcement learning (RL) agent, whose model learns how EM field evolves according to the change of the metagrating structure. The objective of an RL agent here is to find the metagrating structure that yields high deflection efficiency, by *flipping* the material of a cell between silicon and air. Here, MFS is set to 1, i.e., there is no MFS constraint.

**Problem setup.** Here an RL agent undergoes fully-observable Markov decision process described as sequence of tuples $\{s_t, a_t, r_t, s_{t+1}\}_{t=1}^{T}$, where the next state $s_{t+1}$ is determined by the dynamics model $p(s_{t+1} \mid s_t, a_t)$ and the action is the index of cell to flip, $a_t \in \{0, 1, ..., k-1\}$. The state $s_t$ and the reward $r_t$ depend on which RL algorithm is used, and will be defined shortly after. Throughout the decision process, the agent learns to flip cells that maximizes the deflection efficiency $\eta_t$. For training purpose, we implemented Gymnasium (Towers et al., 2023) environment called `deflector-gym`, which is built on top of `meent`. Given an input action $a_t$, the environment modifies current structure $g_t$ and outputs FoMs such as deflection efficiency $\eta_t$ or electric field $v_t$ in deterministic manner.

**Model-based RL vs Model-free RL.** One way to categorize RL algorithms is whether it has an explicit dynamics model $p \approx p_\theta(s_{t+1} \mid s_t, a_t)$, where $p_\theta$ is some neural network. Model-based RL (MBRL) agent utilizes the model $p_\theta$ to produce simulated experiences, from which the policy is improved (Sutton, 1991; Sutton & Barto, 2018). Therefore, MBRL agent is considered to be more *sample efficient* than model-free agent, i.e., requires less interactions with actual environment to train.

For an MBRL algorithm, we chose DreamerV3 (Hafner et al., 2023), the first algorithm that solved `ObtainDiamond` task of MineRL (Guss et al., 2019). DreamerV3 is based on recurrent state-space model (RSSM) (Hafner et al., 2019b) that models dynamics in the latent space. Developed to be robust across varying scales of observations and rewards in different tasks, it has successfully addressed numerous challenges using a single set of hyperparameters, most of which were reused here as well.

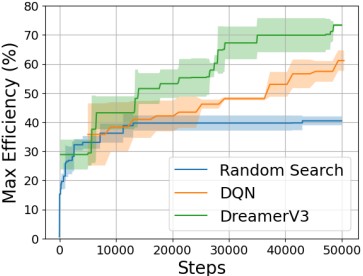

Figure 6: **Learning curve across 3 random seeds.** Historical maximum efficiencies during training phase are plotted.

We compare this with Deep Q Network (DQN) (Mnih et al., 2015), a model-free algorithm, adapted from (Park et al., 2024).

Same as in (Park et al., 2024), our DQN agent receives the reward $r_t = \eta_t - \eta_{t-1}$ and observes the grating pattern as the state $s_t = g_t$. On the other hand, DreamerV3 agent receives reward $r_t = \eta_t$ and observes the grating pattern along with the electric field, $s_t = (g_t, v_t)$, to enable the dynamics model $p_\theta$ to learn underlying physics of the transitions of electric fields. For further training details and the motivation behind the aforementioned reward engineering, we refer readers to Appendix E.1

As was the case in other tasks, DreamerV3 agent showed improved sample efficiency in our task as seen in Figure 6 when compared to DQN. Not only does it shows more effective optimizations at the same training steps, but it also achieves higher maximum deflection efficiency. As a side remark, we mention that the training of RL agents can be massively accelerated with running `meent` in parallel with Ray/RLlib (Liang et al., 2018). Simple comparison of training speed between single worker and multiple workers are shown in Appendix E.1.

**Electric field prediction of the dynamics model.** In addition to the sample efficiency, another advantage of using MBRL in this task is that the dynamics model $p_\theta(s_{t+1} \mid s_t, a_t)$ can be used to predict the electric field of the next state, which not only makes the dynamics model interpretable but also suggests another way of developing an EM surrogate solver in addition to neural operators.

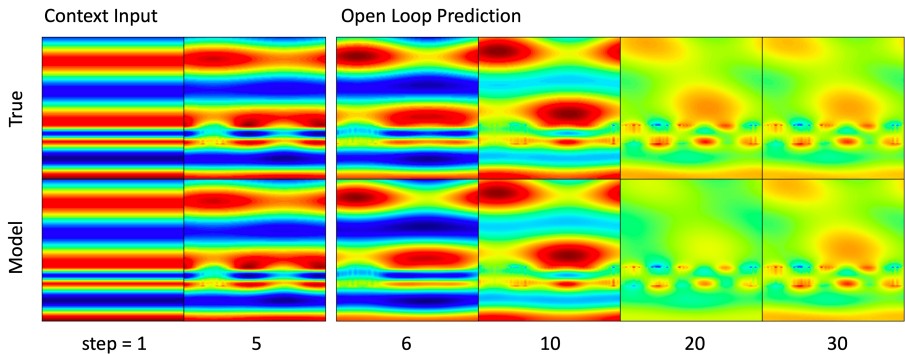

Figure 7: **Comparison between ground truth and prediction.** Rollout trajectory of electric fields showing the ground truth (top) and the predictions from the MBRL dynamics model (bottom). Given previous electric fields from step 1 to 5, the model predicts 25 future electric fields with actions the agent has actually taken, but without any access to ground truth electric field. All of the images are real part of the field.

Figure 7 shows the prediction of the model compared to the ground truth calculated from `meent`. One interesting observation is that, even when the prediction at a certain time step deviates from the ground truth, the model does not compound the error but is able to correct itself to converge to the correct estimation. The robustness of the prediction of the dynamics model is also illustrated in Appendix E.2, where the dynamics model was able to reproduce the correct electric field configuration even for a difficult problem that a neural operator fails to estimate correctly.

### 4.3 Inverse problem: OCD metrology

A semiconductor device is a three-dimensional stack of layers, rendering direct measurement of parameters beneath the surface unfeasible without causing damage. OCD offers a solution to this challenge by redirecting the observation target from the dimensions of physical device to its spectral characteristics (spectrum). Consequently, OCD becomes an inverse problem: we deduce the dimensions of the structure in real space, which are the causal factors of spectrum shape, from observations. The solution involves a probabilistic and iterative process known as spectrum fitting. This necessitates optimization in continuous space, which can be achieved using `meent` with vector modeling.

**Spectrum fitting.** Figure 8 shows the process of finding solution using spectrum fitting. The goal is to estimate $P$ using $S$, as direct observation of $P$ is impossible. To achieve this, we initially create a virtual structure with limited prior knowledge provided by domain experts, and sample the initial parameters $\hat{P}_0$ from a suitable distribution. Subsequently, we generate $\hat{S}_0$ from $\hat{P}_0$ through simulation.

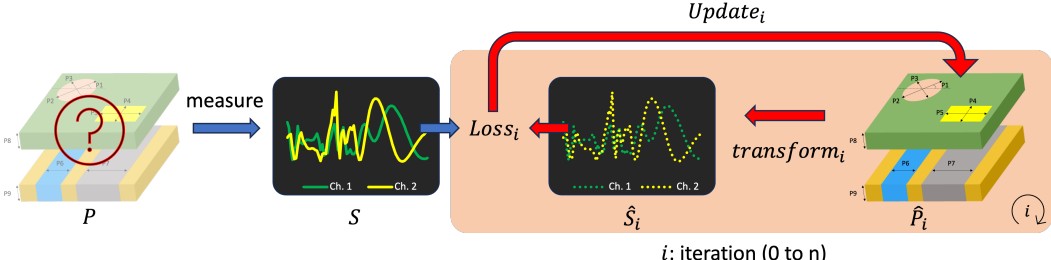

Figure 8: **Schematic diagram of spectrum fitting.** This diagram illustrates the key components of the approach, including the vector of design parameters ($P$) from the real device, the set of spectra ($S$) derived from these parameters, the synthesized spectra at iteration $i$ ($\hat{S}_i$), and the estimated design parameters at the corresponding iteration ($\hat{P}_i$). The methodology involves assessing the distance between $\hat{S}_i$ and $S$ using a loss function, followed by a update of $\hat{P}_i$ to minimize this distance during each iteration.

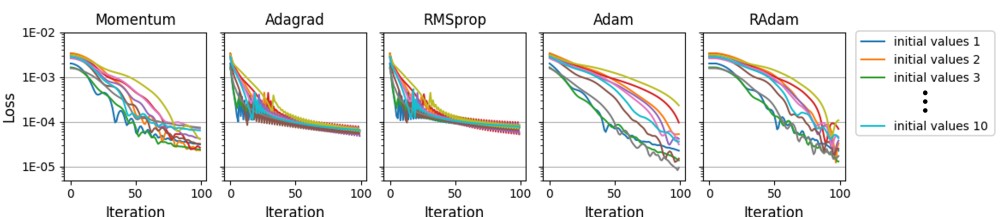

Figure 9: **Loss curves of various gradient-based algorithms for spectrum fitting.** Each plot illustrates the change of loss over iterations, with the y axis represented in logarithmic scale.

We then employ a distance metric as a loss function to quantify the discrepancy between $S$ and $\hat{S}_0$ to compare $P$ and $\hat{P}_0$. The process is followed by backpropagation, which computes gradients of the distance with respect to each element of $\hat{P}_0$. Following that, $\hat{P}_0$ is updated to $\hat{P}_1$. This iterative process can be generalized using $\hat{P}_i$ and $\hat{S}_i$, where $i$ denotes the iteration number. As iterations progress, $\hat{S}_i$ gradually converges towards $S$, and it is expected that $\hat{P}_i$ will similarly approach the parameter set we seek to obtain, $P$.

**Demonstration.** We now pivot our approach to `meent`. Rather than seeking $P$, we utilize `meent` to observe the behavior of optimization algorithms, a subject of keen interest for ML researchers. As an example, we introduce a case study involving eight design parameters with the details provided in Appendix F. Employing spectrum fitting, we present the optimization curve of five distinct gradient-based algorithms: Momentum, Adagrad (Duchi et al., 2011), RMSProp (Hinton et al.), Adam (Kingma & Ba, 2017) and RAdam (Liu et al., 2020). All algorithms share identical $\hat{P}_0$ to ensure consistency, and evaluated repeatedly with 10 different initial conditions to mitigate the randomness associated with initial point location, a critical factor in local optimization algorithms. The purpose of this section is to demonstrate the capabilities of `meent`. The introduction of novel algorithms or the achievement of precise predictions is beyond the scope of this case study.

## 5 CONCLUSION

In our work, we introduce `meent`, a full-wave, differentiable electromagnetic simulation framework. Through its capability for vector-type modeling and automatic differentiation, `meent` operates seamlessly within a continuous space, overcoming the limitations inherent in raster-type modeling for geometry representation. We demonstrate with examples of applications how to use `meent` as a valuable tool to generate data for ML as well as a comprehensive solver for inverse problems, expanding its role beyond that of a simple electromagnetic simulator. This versatility makes `meent` an invaluable framework to both machine learning and optics.

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

# APPENDIX

## CONTENTS

## A    A BRIEF INTRODUCTION TO RCWA

RCWA is based on Faraday's law and Ampére's law of Maxwell's equations (Kim et al., 2012),

$$\nabla \times \mathbf{E} = -j\omega\mu_0\mathbf{H}, \qquad \nabla \times \mathbf{H} = j\omega\varepsilon_0\varepsilon_r\mathbf{E}, \tag{5}$$

where $\mathbf{E}$ and $\mathbf{H}$ are electric and magnetic field in real space, $j$ denotes the imaginary unit number, i.e. $j^2 = -1$, $\omega$ denotes the angular frequency, $\mu_0$ is vacuum permeability, $\varepsilon_0$ is vacuum permittivity, and $\varepsilon_r$ is relative permittivity. After Fourier transform, $\mathbf{E}$ and $\mathbf{H}$ in real space become $\mathfrak{S}$ and $\mathfrak{U}$ in Fourier space, respectively, and Equation 5 then turns into

$$(-j\tilde{k}_z)\mathfrak{S} = \mathbf{\Omega}_L\mathfrak{U}, \qquad (-j\tilde{k}_z)\mathfrak{U} = \mathbf{\Omega}_R\mathfrak{S}, \tag{6}$$

where the matrices $\mathbf{\Omega}_L$ and $\mathbf{\Omega}_R$ are composed of wavevector matrices and convolution matrices that lie in Fourier space, $\tilde{k}_z$ is a normalized wavevector in the $z$-direction. We can find $\mathfrak{S}$ by merging Equation 6 in a single matrix equation as

$$(-j\tilde{k}_z)^2\mathfrak{S} = \mathbf{\Omega}_{LR}^2\mathfrak{S}, \tag{7}$$

where the matrix $\mathbf{\Omega}_{LR}^2$ is a matrix product of $\mathbf{\Omega}_L$ and $\mathbf{\Omega}_R$. As the form implies, this equation can be solved by eigendecomposition of $\mathbf{\Omega}_{LR}^2$ to obtain the eigenvectors $\mathfrak{S}$ and the eigenvalues $(-j\tilde{k}_z)$. Then by substituting the eigenvectors for $\mathfrak{S}$ in Equation 6, the corresponding solution of $\mathfrak{U}$ can be obtained.

$\mathfrak{S}$ and $\mathfrak{U}$, which we have just computed, represent the set of electromagnetic modes (field representation in Fourier space) within a given medium. To understand their interaction with incoming and outgoing light, we employ boundary conditions to ascertain their respective weights of the modes, or in other words, the coefficients in linear combination of the modes. These coefficients describe the extent of each mode's influence on the overall field distribution. Notably, coefficients at the input and output interfaces are designated as diffraction efficiencies, also called the reflectance and transmittance, serving as the primary purpose of RCWA. Subsequently, the inverse transformation from Fourier space to real space enables the reconstruction of the field distribution.

## B    COMPUTING RESOURCES

Table 5: Hardware specification

|         | CPU                  | clock   | # threads | GPU                |
|---------|----------------------|---------|-----------|--------------------|
| Alpha   | Intel Xeon Gold 6138 | 2.00GHz | 80        | TITAN RTX          |
| Beta    | Intel Xeon E5-2650 v4| 2.20GHz | 48        | GeForce RTX 2080ti |
| Gamma   | Intel Xeon Gold 6226R| 2.90GHz | 64        | GeForce RTX 3090   |
| Softmax | Intel i9-13900K      | 3.00GHz | 32        | GeForce RTX 4090   |

## C   CODE SNIPPETS FOR FOMS

**Parameters of Code 1, 2**

- `pattern_input`: The grating pattern $G$
- `wavelength`: The wavelength of light $\lambda$
- `fourier_order`: The Fourier truncation order of RCWA
- `deflected_angle`: The desired deflection angle $\theta$
- `field_res`: The resolution of the field

Please refer to Appendix G for more physical conditions in `meent`.

```python
def get_field(
        pattern_input,
        wavelength=1100,
        deflected_angle=70,
        fourier_order=40,
        field_res=(256, 1, 32)
):
    period = [abs(wavelength / np.sin(deflected_angle / 180 * np.pi))]
    n_ridge = 'p_si__real'
    n_groove = 1
    wavelength = np.array([wavelength])
    grating_type = 0
    thickness = [325] * 8

    if type(n_ridge) is str:
        mat_table = read_material_table()
        n_ridge = find_nk_index(n_ridge, mat_table, wavelength)
    ucell = np.array([[pattern_input]])
    ucell = (ucell + 1) / 2
    ucell = ucell * (n_ridge - n_groove) + n_groove
    ucell_new = np.ones((len(thickness), 1, ucell.shape[-1]))
    ucell_new[0:2] = 1.45
    ucell_new[2] = ucell

    mee = meent.call_mee(
        mode=0, wavelength=wavelength, period=period, grating_type=0, n_I
=1.45, n_II=1.,
        theta=0, phi=0, psi=0, fourier_order=fourier_order, pol=1,
        thickness=thickness,
        ucell=ucell_new
    )
    de_ri, de_ti, field_cell = mee.conv_solve_field(
        res_x=field_res[0], res_y=field_res[1], res_z=field_res[2],
    )
    field_ex = np.flipud(field_cell[:, 0, :, 1])

    return field_ex
```

Code 1: Method for caculating electric field $v$

```
1  def get_efficiency(
2          pattern_input,
3          wavelength=1100,
4          deflected_angle=70,
5          fourier_order=40
6  ):
7
8      period = [abs(wavelength / np.sin(deflected_angle / 180 * np.pi))]
9      n_ridge = 'p_si__real'
10     n_groove = 1
11     wavelength = torch.tensor([wavelength])
12     grating_type = 0
13     thickness = [325]
14
15     if type(n_ridge) is str:
16         mat_table = read_material_table()
17         n_ridge = find_nk_index(n_ridge, mat_table, wavelength)
18     ucell = torch.tensor(np.array([[pattern_input]]))
19     ucell = (ucell + 1) / 2
20     ucell = ucell * (n_ridge - n_groove) + n_groove
21
22     mee = meent.call_mee(
23         backend=2, wavelength=wavelength, period=period, grating_type=0,
       n_I=1.45, n_II=1.,
24         theta=0, phi=0, psi=0, fourier_order=fourier_order, pol=1,
25         thickness=thickness,
26         ucell=ucell
27     )
28     de_ri, de_ti = mee.conv_solve()
29     rayleigh_r = mee.rayleigh_r
30     rayleigh_t = mee.rayleigh_t
31
32     if grating_type == 0:
33         center = de_ti.shape[0] // 2
34         de_ri_cut = de_ri[center - 1:center + 2]
35         de_ti_cut = de_ti[center - 1:center + 2]
36         de_ti_interest = de_ti[center+1]
37
38     else:
39         x_c, y_c = np.array(de_ti.shape) // 2
40         de_ri_cut = de_ri[x_c - 1:x_c + 2, y_c - 1:y_c + 2]
41         de_ti_cut = de_ti[x_c - 1:x_c + 2, y_c - 1:y_c + 2]
42         de_ti_interest = de_ti[x_c+1, y_c]
43
44     return float(de_ti_interest)
```

Code 2: Method for caculating deflection efficiency $\eta$

# D  NEURAL PDE SOLVER

## D.1  TRAINING DETAILS

**Dataset**  We split 10,000 pairs of $(u, v)$ into 8000 training pairs and 2000 test pairs for each of nine physical conditions. An instance of $u$ is sized $1 \times 256 \times 256$, each element indicating whether it is filled or empty. $v$ is sized $2 \times 256 \times 256$, one channel for real part and another for imaginary part of electric field, each element expressing the intensity of electric field. The set of nine physical conditions are shown in the first column of Table 9, and the set is followed from (Seo et al., 2021). Fourier truncation order is set to 40.

For testing zero-shot super-resolution, the structures are transferred to higher resolutions, and corresponding electric fields are calculated with Code 1. Please refer to our GitHub repository for the script to generate the data.

Table 6: Common hyperparameters

| Name | Value |
|------|-------|
| # of Epochs | 100 |
| Optimizer | AdamW |
| Learning rate | 1E-3 |
| LR scheduler | OneCycleLR |
| Base momentum | 0.85 |
| Max momentum | 0.95 |
| Activation | GELU |

**Fourier neural operator (FNO)**  We used 3,268,062 parameters for training FNO. To serve as a baseline, we adhered closely to the architecture described in (Augenstein et al., 2023), except for Tucker factorization.

Table 7: FNO hyperparameters

| Name | Value |
|------|-------|
| # of modes | [24, 24] |
| Lifting channels | 32 |
| Hidden channels | 32 |
| Projection channels | 32 |
| # of layers | 10 |
| Domain padding | 0.015625 |
| Factorization | Tucker |
| Factorization rank | 0.5 |
| Normalization | BatchNorm |

**UNet**  We used vanilla UNet described in the original paper (Ronneberger et al., 2015), of which parameters counts up to 31,036,546.

**Computational resource**  Both FNO and UNet was trained on Beta server of Table 5. FNO was trained for 3.80 hours, and UNet was trained for 1.86 hours. Both algorithms used single GPU of Beta server and consumed most of the GPU memory.

**Remark**  We utilized the FNO code of the original author, under MIT license. Also, the widely used UNet implementation available under the GPL-3.0 license.

## D.2 ABLATION STUDY

We train FNO on various losses shown in Table 8, and name it as {Model}-{Training loss}, e.g. FNO-L2 is a FNO trained with L2 loss. A model is trained specifically for single physical condition.

On metagrating, more intense and complex interactions occur around the grating area. What makes this area more important is that, theoretically, the deflection efficiency can be calculated just with the field profile of grating area. Take a look at the supplementary of (Chen et al., 2022).

Therefore, we derive a simple loss coined as region-wise (RW) L2 loss, which puts more weight on the grating area, s.t. $c_1 + c_2 = 1$. We set $c_1 = 0.7$ and $c_2 = 0.3$. Lastly, H1 loss is a norm in Sobolev space which integrates the norm of first derivative of the target. Training with H1 loss promotes smoother function (Son et al., 2021).

### D.2.1 LOSS FUNCTIONS

Table 8: Loss functions. $\| \cdot \|$ is the Euclidean norm, RW is shorthand for region-wise. $grating, air, glass$ refers to the sets of indices for each region on matrix representation. All losses are relative error, i.e. normalized by the magnitude of the ground truth $y$.

| Name | Notation | Definition |
|---|---|---|
| L2 loss | $L_2$ | $\|y - \hat{y}\| \, / \, \|y\|$ |
| RW L2 loss | $L_{2,rw}$ | $c_1 \cdot L_{2,grating} + c_2 \cdot (L_{2,air} + L_{2,glass})$ |
| H1 loss | $H_1$ | $\sqrt{\left(\|y - \hat{y}\|^2 + \|y' - \hat{y}'\|^2\right) / \left(\|y\|^2 + \|y'\|^2\right)}$ |

Table 9 shows that FNO-RW L2 achieves slightly lower error than FNO-L2, but it is not significant enhancement compared to FNO-H1. FNO-H1 shows best performance across all test metrics, L2, RW L2, and H1. UNet is trained only with L2 loss, serving as a simple baseline. Comparing mean L2 values, 8.71 of FNO-L2 is 76% lower error than 34.80 of UNet-L2.

Table 9: Test error across loss functions. Of the column names, top row is the name of the models and bottom row is the test metrics.

| Condition ($\lambda / \theta$) | UNet-L2 | | | FNO-L2 | | | FNO-RW L2 | | | FNO-H1 | | |
|---|---|---|---|---|---|---|---|---|---|---|---|---|
| | L2↓ | RW L2↓ | H1↓ | L2 | RW L2 | H1 | L2 | RW L2 | H1 | L2 | RW L2 | H1 |
| $1100nm$ / $70°$ | 34.04 | 22.64 | 33.28 | 7.15 | 6.52 | 14.57 | 7.35 | 4.14 | 10.95 | 6.04 | 3.56 | 6.35 |
| $1100nm$ / $60°$ | 41.61 | 41.86 | 47.82 | 14.57 | 17.37 | 26.65 | 16.03 | 14.7 | 24.11 | 11.09 | 10.98 | 14.62 |
| $1100nm$ / $50°$ | 24.37 | 56.33 | 61.05 | 2.52 | 22.38 | 33.70 | 2.58 | 21.93 | 33.81 | 2.07 | 12.33 | 17.37 |
| $1000nm$ / $70°$ | 43.44 | 22.17 | 29.55 | 15.15 | 5.7 | 12.16 | 15.19 | 4.93 | 11.91 | 9.02 | 3.35 | 5.42 |
| $1000nm$ / $60°$ | 34.02 | 54.74 | 56.98 | 10.7 | 21.89 | 32.66 | 9.5 | 22.93 | 32.74 | 7.88 | 15.05 | 19.21 |
| $1000nm$ / $50°$ | 28.46 | 39.62 | 44.28 | 2.88 | 12.34 | 22.51 | 2.25 | 11.66 | 21.50 | 2.19 | 8.26 | 12.15 |
| $900nm$ / $70°$ | 40.78 | 27.21 | 34.25 | 15.14 | 8.37 | 15.05 | 13.63 | 6.51 | 12.67 | 10.8 | 5.03 | 7.31 |
| $900nm$ / $60°$ | 31.36 | 30.53 | 34.07 | 6.07 | 11.10 | 17.27 | 5.47 | 9.08 | 14.61 | 4.85 | 7.26 | 9.24 |
| $900nm$ / $50°$ | 35.11 | 51.64 | 51.59 | 4.23 | 22.87 | 30.79 | 3.77 | 19.89 | 27.33 | 3.29 | 14.91 | 17.75 |
| Mean | 34.80 | 38.53 | 43.65 | 8.71 | 14.28 | 22.81 | 8.42 | 12.86 | 21.07 | **6.36** | **8.97** | **12.16** |
| ±Std | ±5.95 | ±12.81 | ±10.79 | ±5.95 | ±6.58 | ±7.93 | ±5.12 | ±6.92 | ±8.47 | **±3.32** | **±4.31** | **±5.00** |

### D.2.2 DEVICE REPRESENTATIONS

We carried out experiments on three types of representation, each with simple motivation. Model trained on refractive index matrix showed best result. Three representations are as follows: (a) Binary matrix. Simply filled with -1, 1 to distinguish a material and the other, only the pattern of grating area varies across the data. This representation requires no knowledge of physics, such as refractive index. Assuming the model learns underlying physics of the electromagnetism, physicals features such as refractive index is implicitly distilled in the model. (b) Categorical matrix. More general representation than the binary matrix, where a device consists of three materials. This requires larger space complexity since, each element need to be one-hot encoded. (c) Refractive index matrix. This representation is more intuitive in optics perspective since it directly models the device with important optical properties. The elements are set with `meent`'s refractive index table, and the missing conditions are interpolated. The kind of material and wavelength determines the refractive index.

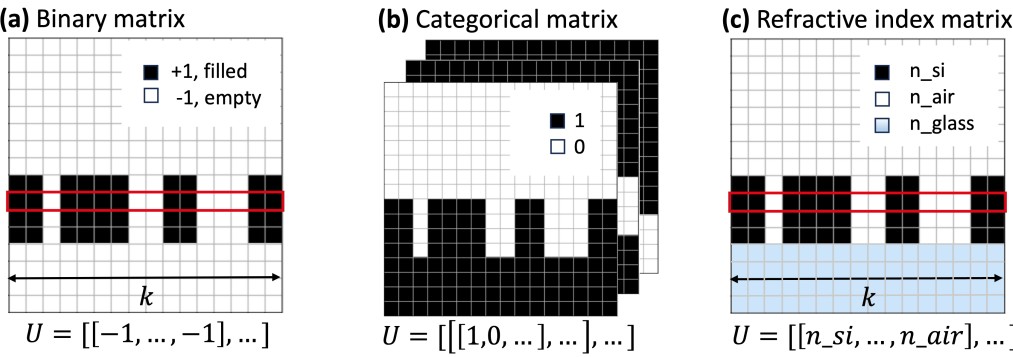

Figure 10: **Three types of representations.**

Table 10: Mean test L2 error across representations. Only FNO-H1, which showed best result, is tested against different representations. Averaged along all nine conditions.

| | FNO-H1 | | |
| --- | --- | --- | --- |
| | Binary | Categorical | Refractive index |
| Mean L2↓ | 6.36 | 4.27 | **4.17** |
| ±Std | ±3.32 | ±1.87 | ±**1.79** |

Based on the experimental results, we concluded that, although the air and glass areas of the device remain constant across all data, it is crucial to encode this information along with the grating area. This is because it is important to signal to the model that the interactions between the grating and air, as well as the grating and glass, are distinct and intricately intertwined.

# E METASURFACE DESIGN

## E.1 TRAINING RL AGENT

Budget refers to the total episode steps consumed for training an agent.

We limit the budget to 50,000 steps, which is 75% less than the budget used in (Park et al., 2024).

Table 11: Fixed configurations for all algorithms

| Parameter | Value |
|---|---|
| Budget | 50,000 steps |
| Initial structure | $g_0 = [1, 1, \dots, 1]$ |
| Episode length | $T = 512$ |
| Replay buffer size | 25,000 (adjusted proportionally to budget) |
| Asynchronous environments | 8 |
| Number of cells | $k = 256$ |
| Wavelength | $\lambda = 1100\,\text{nm}$ |
| Desired deflection angle | $\theta = 70°$ |
| Fourier truncation order | 40 |

**Dataset**  Please refer to out Github repository for RL environment utilizing `meent`.

**DQN**  We mostly follow the previous work (Park et al., 2024). The structure $g_t$ is encoded with shallow UNet, and the reward $r_t = \eta_t - \eta_{t-1}$ is received. For fair comparison with DreamerV3 L (Hafner et al., 2023), following details were changed. Much larger number of parameters (70,711,873) was used, and physics-informed weight initialization was replaced by Pytorch's default initialization (Ansel et al., 2024). Additionally, 1000 steps were used for warmup to fill empty replay buffer.

**DreamerV3**  DreamerV3 was trained with 99,789,440 parameters. Most of the hyperparameters from original paper (Hafner et al., 2023) were reused, excluding: 1,024 steps were used for warmup, replay ratio was increased for sample efficiency, batch size and sequence length was adjusted due to our task's relatively shorter episode length.

Table 12: DreamerV3 hyperparameters

| Name | Value |
|---|---|
| Model size | L |
| Replay ratio | 2 |
| Batch size | 8 |
| Sequence length | 32 |

As mentioned in the main text, DreamerV3 agent observes additional feature, the electric field $v_t$. The structure $g_t$ and electric field $v_t$ are encoded by MLP and CNN respectively, and concatenated to form a latent state. With the input action and latent state, the dynamics model predicts next state, reward and done condition. Simply put, the dynamics model functions as the environment.

Emprically, DreamerV3 failed the metasurface optimization with the reward of efficiency change $r_t = \eta_t - \eta_{t-1}$. From the experimental observation, we hypothesized that if the model truly understands the underlying physics, the reward predictor should directly predict efficiency $r_t = \eta_t$, not the change of efficiency. This hypothesis was inspired by the fact that efficiency can be derived from electric field as mentioned in D.2. With this hypothesized reward engineering, DreamerV3 agent successfully learnt to optimize the metasurface structure.

**Computational resource**  For servers in Table 5, DreamerV3 was trained for 10.88 hours on Softmax server with single GPU. DQN was trained for 1.6 hours on Alpha server. Both algorithm used its server's single GPU and consumed most of the GPU memory.

Despite the big difference in training time, when the device is expanded to high dimensionality, the simulation time can occupy the biggest portion of training time (Augenstein et al., 2023). The main point of our example here is to show the dynamics model's potential as surrogate solver in decision process, and we leave high dimensional problem as a future work.

**Parallelization**  Example benchmark on the axis of number of workers. The code for adapting RLlib wil be provided on our Github repository. Figure 11 shows that the calculation time sub-linearly decreases as the number of workers increases.

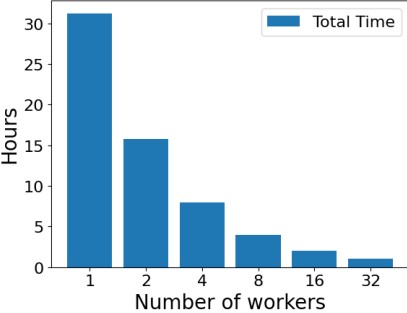

Figure 11: **Parallelization of `meent` with Ray/RLlib.**

**Remark**  The experimental code was adapted from SheepRL (EclecticSheep et al., 2023) for DreamerV3. We utilized the previous version of DreamerV3, prior to the updated release on April 17, 2024. Details of the training procedure and architecture are beyond the scope of this paper. For comprehensive information, please refer to (Hafner et al., 2023).

## E.2  HIGH IMPACT CELL

We spotted the accurate prediction of dynamics model for scarcely happening transition. Introduced in (Seo et al., 2021), a high impact cell refers to a cell that incurs abrupt change in some FoMs when flipped, which is very small change in the material distribution. To artificially create this case, a fully trained agent is run an episode and produces a trajectory. At the step of the trajectory when the efficiency reached about 75%, a high impact cell is manually found by flipping every cell of the structure at the step. With the found index of high impact cell, the action is fed into dynamics model, to predict next electric field.

As shown in Figure 12, dynamics model accurately captures the transition whereas FNO-H1 model, trained under same physical condition in Table 9, entirely fails to predict this phenomena. FNO-H1 might perform better if trained with similar distributions of data, but it is very difficult to draw similar patterns from extremely large design space size, $2^{256}/256$, where the denominator 256 is for the periodicity.

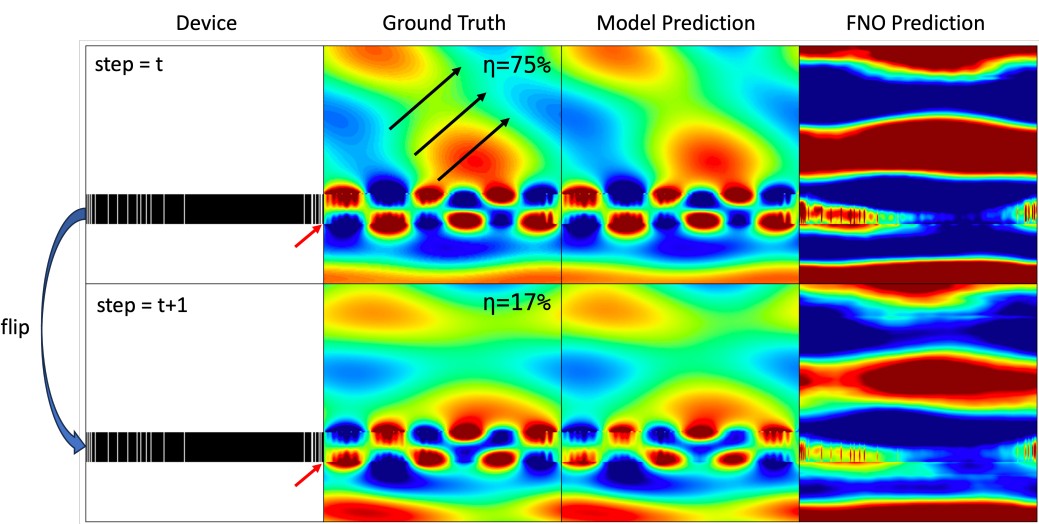

Figure 12: **High impact cell phenomena.** Flipping a single $254^{th}$ cell from silicon to air (red arrow) results in completely different electric field and large decrease in deflection efficiency $\eta$ from about 75% to 17%. Our world model is able to capture the transition. Field intensity is clipped from 0.4 to 0.6 for clearer visualization. All of images are real part of the field.

## F OCD DEMONSTRATION

To simulate a real-world scenario where we have the real devices and their spectra, we first determine values for ground truth of the design parameters denoted as $P$, and generate spectra $S$ with simulation. These values are typically provided from domain experts. Our chosen values are in Table 13.

Table 13: Design parameter information

| Parameter | Variable name | Mean | STD | Ground Truth |
|-----------|---------------|------|-----|--------------|
| P1 | l1_o1_length_x | 100 | 3 | 101.5 |
| P2 | l1_o1_length_y | 80 | 3 | 81.5 |
| P3 | l1_o2_length_x | 100 | 3 | 98.5 |
| P4 | l1_o2_length_y | 80 | 3 | 81.5 |
| P5 | l2_o1_length_x | 30 | 2 | 31 |
| P6 | l2_o2_length_x | 50 | 1 | 49.5 |
| P7 | l1_thickness | 200 | 10 | 205 |
| P8 | l2_thickness | 300 | 10 | 305 |

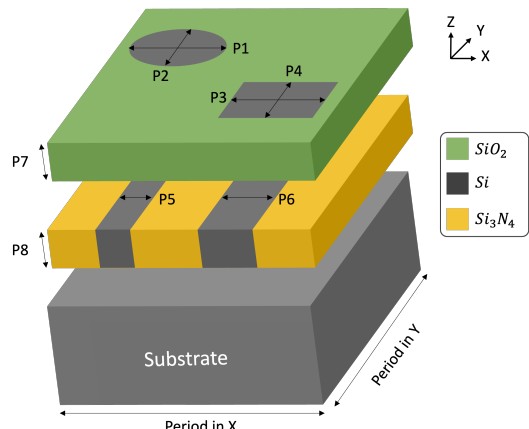

Figure 13: **Stack in experiment.**

Figure 13 depicts the stack utilized in the demonstration. Two layers are stacked on the silicon substrate, each containing objects within. Light is illuminated from the top, and the reflected light is acquired and processed into spectra.

To address this inverse problem of finding design parameters from spectra, initial values for optimization need to be determined. These conditions are also provided by domain experts. In this demonstration, these values are drawn from a normal distribution without correlation. The mean and standard deviation (STD) are listed in Table 13.

The hyperparameters utilized for the optimization demonstration are presented in Table 14. Default values from PyTorch are used for conditions not explicitly mentioned. The learning rate was determined through a concise parameter-sweep test, which assessed three different values of the learning rate for each optimizer, as presented in Figure 14.

Table 14: Design parameter information

| Optimizer | Learning Rate | Other conditions |
|-----------|---------------|------------------|
| Momentum | 1E2 | momentum: 0.9 |
| Adagrad | 1E0 | |
| RMSProp | 1E-1 | |
| Adam | 1E-1 | |
| RAdam | 1E0 | |

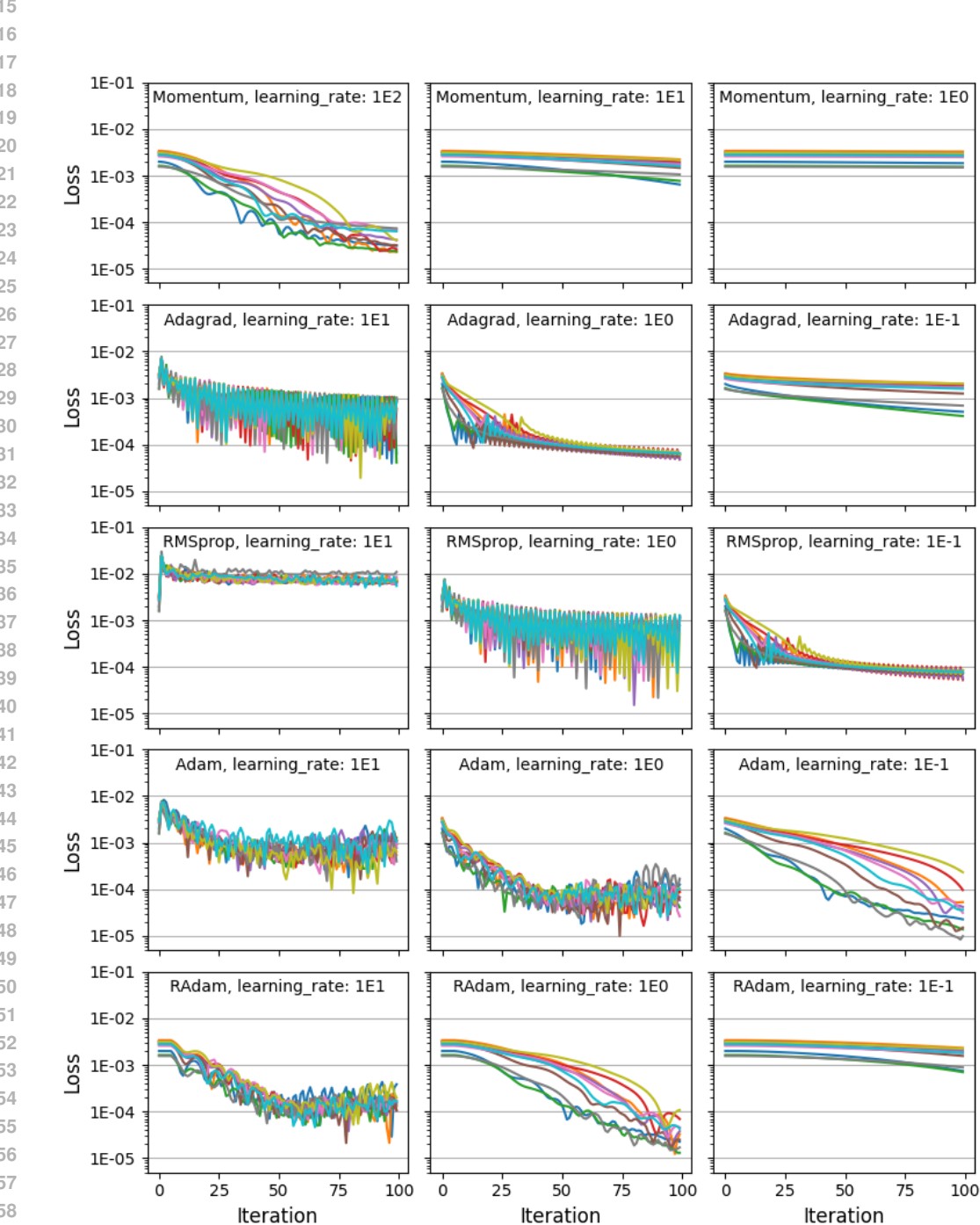

Figure 14: **Hyperparameter sweep.**

## G    BACKGROUND THEORY

RCWA is the sequence of the following processes: solving the Maxwell's equations, finding the eigenmodes of a layer and connecting these layers including the superstrate and substrate to calculate the diffraction efficiencies. Precisely, the electromagnetic field and permittivity geometry are transformed from the real space to the Fourier space (also called the reciprocal space or k-space) by Fourier analysis. Maxwell's equations are then solved per layer through convolution operation, and a general solution of the field in each direction can be obtained. This general solution can be represented in terms of eigenmodes (eigenvectors) and eigenvalues with eigendecomposition, and used to calculate diffraction efficiencies by applying boundary conditions and connecting to adjacent layers.

This chapter provides a comprehensive explanation of the theories, formulations and implementations of `meent` in the following sections:

1. Structure design: the device geometry is defined and modeled within `meent` framework.

2. Fourier analysis of geometry: the device geometry is transformed into the Fourier space, allowing the decomposition of the structure into its corresponding spatial frequency components.

3. Eigenmodes identification: RCWA identifies the eigenmodes that present within each layer of the periodic structure. These eigenmodes represent the possible electromagnetic field solutions that can exist within the system.

4. Connecting layers: Rayleigh coefficients and diffraction efficiencies are determined using the transfer matrix method by connecting the layers. This step enables the determination of the overall electromagnetic response of the entire system.

5. Enhanced transmittance matrix method: the implementation technique that avoids the inversion of some matrices which are possibly ill-conditioned.

6. Topological derivative vs Shape derivative: two types of derivatives that `meent` supports are explained.

### G.1    STRUCTURE DESIGN

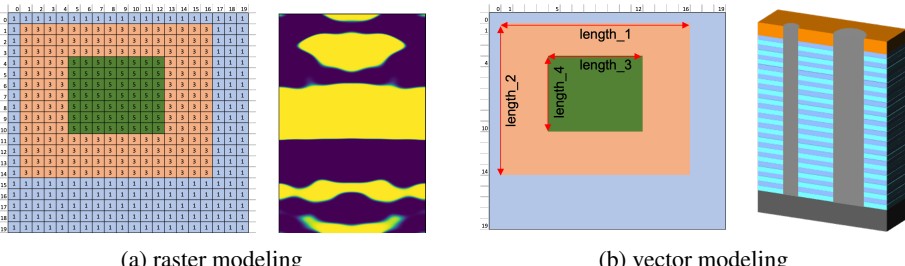

(a) raster modeling                                    (b) vector modeling

Figure 15: **Two types of geometry modeling: raster and vector.** The left of (a) and (b) show how the geometry is formed by each method and the right figures are the representative applications - metasurface design and OCD.

`meent` supports two distinct types of geometry modeling: the raster modeling and the vector modeling. In the raster modeling, the device geometry is gridded and filled with the refractive index of the corresponding material as in Figure 15a. This approach is advantageous for solving optimization problems related to freeform metasurfaces. The vector modeling (shown in Figure 15b), on the other hand, represents the geometry as an union of primitive shapes and each primitive shape is defined by edges and vertices like vector-type image. Consequently, it is memory-efficient and has less parameters to optimize by not keeping the whole array as the raster-type does. This feature is especially valuable in OCD metrology where semiconductor device comprises highly complex structures. raster-type methods may become impractical in such scenarios due to the limitations of grid-based representations. One of the key advantages provided by vector modeling is that the minimum feature size is not constrained by the grid size. This flexibility allows for more accurate and detailed representation of complex structures, making vector modeling essential for accurate simulation.

### G.2    FOURIER ANALYSIS OF GEOMETRY

In RCWA, the device geometry needs to be mapped to the Fourier space using Fourier analysis. To achieve this, the device is sliced into multi-layers so that each layer has Z-invariant (layer stacking direction) permittivity distribution. In other words, the permittivity can be considered as a piecewise-constant function that varies in X

and Y but not Z direction in each layer. Then the geometry in real space can be expressed as a weighted sum of Fourier basis:

$$\varepsilon(x,y) = \sum_{m=-\infty}^{\infty} \sum_{n=-\infty}^{\infty} c_{n,m} \cdot \exp\left[j \cdot 2\pi \left(\frac{x}{\Lambda_x}m + \frac{y}{\Lambda_y}n\right)\right], \tag{8}$$

where $\Lambda_x, \Lambda_y$ are the period of the unit cell and $c_{n,m}$ is the Fourier coefficients ($m^{th}$ in X and $n^{th}$ in Y). However, due to the limitation of the digital computations, this has to be approximated with truncation:

$$\varepsilon(x,y) \simeq \sum_{m=-M}^{M} \sum_{n=-N}^{N} c_{n,m} \cdot \exp\left[j \cdot 2\pi \left(\frac{x}{\Lambda_x}m + \frac{y}{\Lambda_y}n\right)\right], \tag{9}$$

where $M, N$ are the Fourier Truncation Order (FTO, the number of harmonics in use) in the X and Y direction, and these can be considered as hyperparamters that affects the simulation accuracy.

Here, $c_{n,m}$ is the permittivity distribution in the Fourier space which is our interest and can be found by one of these two methods: Discrete Fourier Series (DFS) or Continuous Fourier Series (CFS). To be clear, CFS is Fourier series on piecewise-constant function (permittivity distribution in our case). This name was given to emphasize the characteristics of each type by using opposing words. The output array of DFS and CFS have the same shape and can be substituted for each other.

In DFS, the function $\varepsilon(x, y)$ to be transformed is sampled at a finite number of points, and this means it's given in matrix form with rows and columns, $\varepsilon_{\mathbf{r},\mathbf{c}}$. The coefficients of DFS are then given by this equation:

$$c_{n,m} = \frac{1}{P_x P_y} \sum_{\mathbf{c}=0}^{P_x-1} \sum_{\mathbf{r}=0}^{P_y-1} \varepsilon_{\mathbf{r},\mathbf{c}} \cdot \exp\left[-j \cdot 2\pi \left(\frac{m}{P_x}\mathbf{c} + \frac{n}{P_y}\mathbf{r}\right)\right], \tag{10}$$

where $P_x, P_y$ are the sampling frequency (the size of the array), $\varepsilon_{\mathbf{r},\mathbf{c}}$ is the $(\mathbf{r}, \mathbf{c})^{th}$ element of the permittivity array.

There is an essential but easily overlooked fact: the sampling frequency ($P_x, P_y$) is very important in DFS (Smith, 1999; Antoniou, 2005; Kreyszig et al., 2011). If this is not enough, an aliasing occurs: DFS cannot correctly capture the original signal (you can easily see the wheels of a running car in movies rotating in the opposite direction; this is also an aliasing and called the wagon-wheel effect). In RCWA, this may occur during the process of sampling the permittivity distribution. To resolve this, meent provides a scaling function by default - that is simply to increase the size of the permittivity array by repeatedly replicating the elements while keeping the original shape of the pattern. This option improves the representation of the geometry in the Fourier space and results in more accurate RCWA simulations.

CFS utilizes the entire function to find the coefficients while DFS uses only some of them. This means that CFS prevents potential information loss coming from the intrinsic nature of DFS, thereby enables more accurate simulation. The Fourier coefficients can be expressed as follow:

$$c_{n,m} = \frac{1}{\Lambda_x \Lambda_y} \int_{x_0}^{x_0+\Lambda_x} \int_{y_0}^{y_0+\Lambda_y} \varepsilon(x,y) \cdot \exp\left[-j \cdot 2\pi \left(\frac{m}{\Lambda_x}x + \frac{n}{\Lambda_y}y\right)\right] dy dx. \tag{11}$$

The information that CFS needs are the points of discontinuity and the permittivity value in each area sectioned by those points, whereas DFS needs the whole permittivity array as in Figure 15.

DFS and CFS have its own advantages and one can be chosen according to the purpose of the simulation. Basically, DFS is proper for Raster modeling since its operations are mainly on the pixels (array) and the input of the Raster modeling is the array. This is naturally connected to the pixel-wise operation (cell flipping) in metasurface freeform design. CFS is suitable for Vector modeling because it deals with the graph (discontinuous points and length) of the objects and Vector modeling takes that graph as an input. Hence it enables direct and precise optimization of the design parameters (such as the width of a rectangle) without grid that severely limits the resolution. We will address this in section G.6.

## G.3 EIGENMODES IDENTIFICATION

Once the permittivity distribution is mapped to the Fourier space, the next step is to apply Maxwell's equations to identify the eigenmodes of each layer. In this section, we extend the mathematical formulation of the 1D conical incidence case described in (Moharam et al., 1995a) to the 2D grating case as illustrated in Figure 16. To ensure the consistency and clarity, we adopt the same notations and the sign convention of $(+jwt)$. We consider the normalized excitation wave at the superstrate to take the following form:

$$\mathbf{E}_{inc} = \mathbf{u} \cdot e^{-jk_0 n_I(\sin\theta \cdot \cos\phi \cdot x + \sin\theta \cdot \sin\phi \cdot y + \cos\theta \cdot z)}, \tag{12}$$

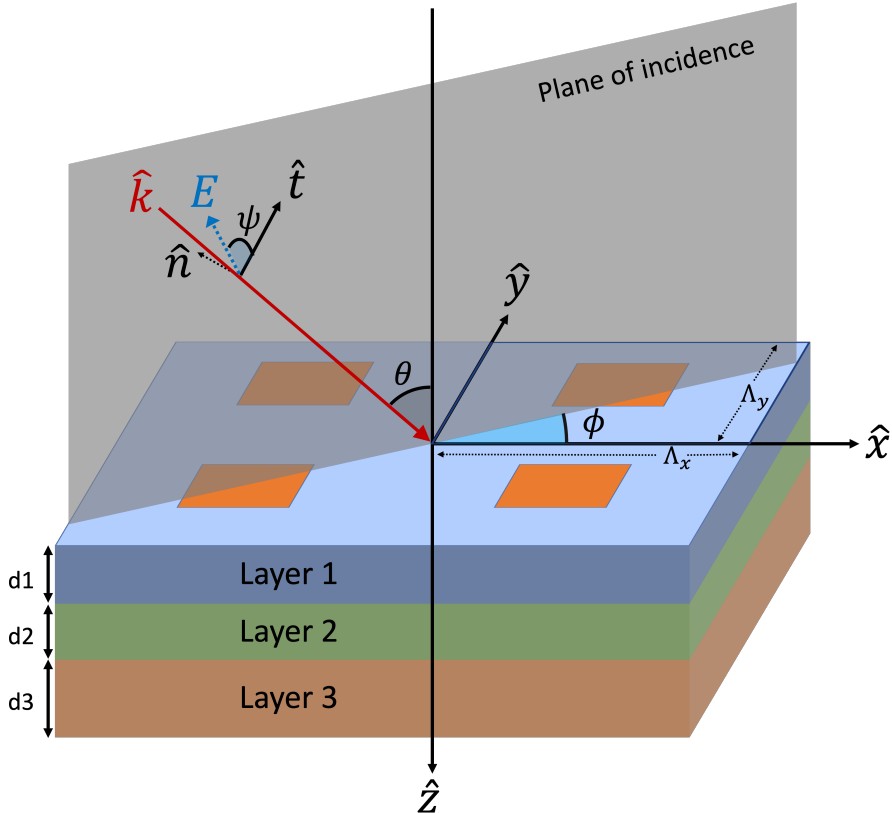

Figure 16: **Geometry for the stack.** Two-dimensional grating layers and incident ray.

where **u** is the normalized amplitudes of the wave in each direction:

$$\mathbf{u} = (\cos\psi \cdot \cos\theta \cdot \cos\phi + \sin\psi \cdot \sin\phi)\hat{x} + (\cos\psi \cdot \cos\theta \cdot \sin\phi + \sin\psi \cdot \cos\phi)\hat{y} + (\cos\psi \cdot \sin\theta)\hat{z}, \tag{13}$$

and $k_0 = 2\pi/\lambda_0$ with $\lambda_0$ the wavelength of the light in free space, $\mathbf{n}_I$ is the refractive index of the superstrate, $\theta$ is the angle of incidence, $\phi$ is the rotation (azimuth) angle and $\psi$ is the angle between the electric field vector and the plane of incidence.

The electric fields in the superstrate and substrate (we will designate these layers by I and II as in (Moharam et al., 1995a)) can be expressed as a sum of incident, reflected and transmitted waves as the Rayleigh expansion (William, 1907; Petit, 1980; Huber et al., 2009):

$$\mathbf{E}_{\mathrm{I}} = \mathbf{E}_{inc} + \sum_{m=-M}^{M} \sum_{n=-N}^{N} \mathbf{R}_{n,m} e^{-j(k_{x,m}x + k_{y,n}y - k_{\mathrm{I},z,(n,m)}z)}, \tag{14}$$

$$\mathbf{E}_{\mathrm{II}} = \sum_{m=-M}^{M} \sum_{n=-N}^{N} \mathbf{T}_{n,m} e^{-j\{k_{x,m}x + k_{y,n}y + k_{\mathrm{II},z,(n,m)}(z-d)\}}, \tag{15}$$

where $M$ and $N$ are the Fourier Truncation Order (FTO) which is related to the number of harmonics in use, and the in-plane components of the wavevector ($k_{x,m}$ and $k_{y,n}$) are determined by the Bloch's theorem (this has many names and one of them is Floquet condition) (Gómez García & Fernández-Álvarez, 2015; Joannopoulos & Steven G. Johnson, 2008),

$$k_{x,m} = k_0\left(\mathbf{n}_\mathrm{I}\sin\theta\cos\phi - m\frac{\lambda_0}{\Lambda_x}\right), \tag{16}$$

$$k_{y,n} = k_0\left(\mathbf{n}_\mathrm{I}\sin\theta\sin\phi - n\frac{\lambda_0}{\Lambda_y}\right), \tag{17}$$

where $\Lambda_x$ and $\Lambda_y$ are the period of the unit cell, and the out-of-plane wavevector is determined from the dispersion relation:

$$k_{\ell,z,(n,m)} = \begin{cases} + [(k_0 \mathbf{n}_\ell)^2 - k_{x,m}{}^2 - k_{y,n}{}^2]^{1/2} & , \quad \text{if} \quad (k_{x,m}{}^2 + k_{y,n}{}^2) < (k_0 \mathbf{n}_\ell)^2 \\ -j[k_{x,m}{}^2 + k_{y,n}{}^2 - (k_0 \mathbf{n}_\ell)^2]^{1/2} & , \quad \text{if} \quad (k_{x,m}{}^2 + k_{y,n}{}^2) > (k_0 \mathbf{n}_\ell)^2 \end{cases}, \quad \ell = \text{I, II.} \quad (18)$$

Here, $k_{\ell,z,(n,m)}$ can be categorized into propagation mode and evanescent mode depending on whether it's real or imaginary. $\mathbf{R}_{n,m}$ and $\mathbf{T}_{n,m}$ are the Rayleigh coefficients (also called the reflection and transmission coefficients): $\mathbf{R}_{n,m}$ is the normalized (3-dimensional) vector of electric field amplitude which is the ($m^{th}$ in X and $n^{th}$ in Y) mode of reflected waves in the superstrate and $\mathbf{T}_{n,m}$ is the normalized (3-dimensional) vector of electric field amplitude which is the ($m^{th}$ in X and $n^{th}$ in Y) mode of transmitted waves in the substrate.

Inside the grating layer, the electromagnetic field can be expressed as a superposition of plane waves by the Bloch's theorem:

$$\mathbf{E}_g(x,y,z) = \sum_{m=-M}^{M} \sum_{n=-N}^{N} \mathfrak{S}_{g,(n,m)} \cdot e^{-j(k_{x,m}x + k_{y,n}y + k_{g,z}z)}, \quad (19)$$

$$\mathbf{H}_g(x,y,z) = \sum_{m=-M}^{M} \sum_{n=-N}^{N} \mathfrak{U}_{g,(n,m)} \cdot e^{-j(k_{x,m}x + k_{y,n}y + k_{g,z}z)}, \quad (20)$$

where $k_{g,z}$ is the wavevector in Z-direction (this is unique per layer hence the notation g was kept to distinguish) and $\mathfrak{S}_{g,(n,m)}$ and $\mathfrak{U}_{g,(n,m)}$ are the vectors of amplitudes in each direction at $(m,n)^{th}$ order:

$$\mathfrak{S}_{g,(n,m)} = \mathfrak{S}_{g,(n,m),x}\,\hat{x} + \mathfrak{S}_{g,(n,m),y}\,\hat{y} + \mathfrak{S}_{g,z}\,\hat{z}, \quad (21)$$

$$\mathfrak{U}_{g,(n,m)} = \mathfrak{U}_{g,(n,m),x}\,\hat{x} + \mathfrak{U}_{g,(n,m),y}\,\hat{y} + \mathfrak{U}_{g,z}\,\hat{z}. \quad (22)$$

It is also possible to detach wavevector term on $z$ from exponent and combine with $\mathfrak{S}_{g,(n,m)}$ and $\mathfrak{U}_{g,(n,m)}$ in Equations 19 and 20 to make $\mathbf{S}_{g,(n,m)}(z)$ and $\mathbf{U}_{g,(n,m)}(z)$ which are dependent on $z$ as shown below:

$$\mathbf{S}_{g,(n,m)}(z) = \mathfrak{S}_{g,(n,m)} \cdot e^{-jk_{g,z}z}, \quad (23)$$

$$\mathbf{U}_{g,(n,m)}(z) = \mathfrak{U}_{g,(n,m)} \cdot e^{-jk_{g,z}z}, \quad (24)$$

then Equations 19 and 20 become

$$\mathbf{E}_g(x,y,z) = \sum_{m=-M}^{M} \sum_{n=-N}^{N} \mathbf{S}_{g,(n,m)}(z) \cdot e^{-j(k_{x,m}x + k_{y,n}y)}, \quad (25)$$

$$\mathbf{H}_g(x,y,z) = \sum_{m=-M}^{M} \sum_{n=-N}^{N} \mathbf{U}_{g,(n,m)}(z) \cdot e^{-j(k_{x,m}x + k_{y,n}y)}. \quad (26)$$

Equations 19 and 20 are used in (Liu & Fan, 2012; Yoon & Rho, 2021; Kim & Lee, 2023) and Equations 25 and 26 in (Moharam et al., 1995a; Rumpf, 2006). Whichever is used, the result is the same: we will show the development using $(\mathfrak{S}_{g,(n,m)}, \mathfrak{U}_{g,(n,m)})$ with the eigendecomposition and then come back to $(\mathbf{S}_{g,(n,m)}(z)$ and $\mathbf{U}_{g,(n,m)}(z))$ with the partial differential equations.

The behavior of the electromagnetic fields can be described by the formulae, called the Maxwell's equations. Among them, we will use the third and fourth equations,

$$\nabla \times \mathbf{E} = -j\omega\mu_0 \mathbf{H}, \quad (27)$$

$$\nabla \times \mathbf{H} = j\omega\varepsilon_0\varepsilon_r \mathbf{E}, \quad (28)$$

to find the electric and magnetic field inside the grating layer - $\mathbf{E}_g$ and $\mathbf{H}_g$. Since RCWA is a technique that solves Maxwell's equations in the Fourier space, curl operator in real space becomes multiplication and multiplication in real space becomes the convolution operator. For this convolution operation, the full set of the modes of the fields and the geometry are required so we introduce a vector notation in the subscript to denote it's a vector with all the harmonics in use, i.e.,

$$\boldsymbol{F}_{g,\vec{r}} = \begin{bmatrix} F_{g,(-N,-M),r} & \cdots & F_{g,(-N,M),r} & F_{g,(-N+1,-M),r} & \cdots & F_{g,(-N+1,M),r} & \cdots & F_{g,(N,M),r} \end{bmatrix}^T, \quad (29)$$

where $\boldsymbol{F} \in \{S, U, \mathfrak{S}, \mathfrak{U}\}$ and $r \in \{x, y, z\}$. Some variables will be scaled by some factors:

$$\tilde{\mathbf{H}}_g = -j\sqrt{\varepsilon_0/\mu_0}\mathbf{H}_g, \quad \tilde{k}_x = k_x/k_0, \quad \tilde{k}_y = k_y/k_0, \quad \tilde{k}_{g,z} = k_{g,z}/k_0, \quad \tilde{z} = k_0 z. \quad (30)$$

Substituting Equations 19 and 20 ($\mathbf{E}_g$ and $\tilde{\mathbf{H}}_g$ with $\mathfrak{S}_g$ and $\mathfrak{U}_g$) into Equations 27 and 28 (Maxwell's equations) and eliminating Z-directional components ($\mathbf{E}_{g,z}$ and $\tilde{\mathbf{H}}_{g,z}$) derive the matrix form of the Maxwell's equations composed of in-plane components ($\hat{x}, \hat{y}$) in the Fourier space:

$$(-j\tilde{k}_{g,z}) \begin{bmatrix} \mathfrak{S}_{g,\vec{x}} \cdot e^{-j\tilde{k}_{g,z}\tilde{z}} \\ \mathfrak{S}_{g,\vec{y}} \cdot e^{-j\tilde{k}_{g,z}\tilde{z}} \end{bmatrix} = \mathbf{\Omega}_{g,L} \begin{bmatrix} \mathfrak{U}_{g,\vec{x}} \cdot e^{-j\tilde{k}_{g,z}\tilde{z}} \\ \mathfrak{U}_{g,\vec{y}} \cdot e^{-j\tilde{k}_{g,z}\tilde{z}} \end{bmatrix} \tag{31}$$

$$(-j\tilde{k}_{g,z}) \begin{bmatrix} \mathfrak{U}_{g,\vec{x}} \cdot e^{-j\tilde{k}_{g,z}\tilde{z}} \\ \mathfrak{U}_{g,\vec{y}} \cdot e^{-j\tilde{k}_{g,z}\tilde{z}} \end{bmatrix} = \mathbf{\Omega}_{g,R} \begin{bmatrix} \mathfrak{S}_{g,\vec{x}} \cdot e^{-j\tilde{k}_{g,z}\tilde{z}} \\ \mathfrak{S}_{g,\vec{y}} \cdot e^{-j\tilde{k}_{g,z}\tilde{z}} \end{bmatrix} \tag{32}$$

$$(-j\tilde{k}_{g,z})^2 \begin{bmatrix} \mathfrak{S}_{g,\vec{x}} \cdot e^{-j\tilde{k}_{g,z}\tilde{z}} \\ \mathfrak{S}_{g,\vec{y}} \cdot e^{-j\tilde{k}_{g,z}\tilde{z}} \end{bmatrix} = \mathbf{\Omega}_{g,LR}^2 \begin{bmatrix} \mathfrak{S}_{g,\vec{x}} \cdot e^{-j\tilde{k}_{g,z}\tilde{z}} \\ \mathfrak{S}_{g,\vec{y}} \cdot e^{-j\tilde{k}_{g,z}\tilde{z}} \end{bmatrix} \tag{33}$$

where

$$\mathbf{\Omega}_{g,L} = \begin{bmatrix} (-\tilde{\mathbf{K}}_x [\![\varepsilon_{r,g}]\!]^{-1}\tilde{\mathbf{K}}_y) & (\tilde{\mathbf{K}}_x [\![\varepsilon_{r,g}]\!]^{-1}\tilde{\mathbf{K}}_x - \mathbf{I}) \\ (\mathbf{I} - \tilde{\mathbf{K}}_y [\![\varepsilon_{r,g}]\!]^{-1}\tilde{\mathbf{K}}_y) & (\tilde{\mathbf{K}}_y [\![\varepsilon_{r,g}]\!]^{-1}\tilde{\mathbf{K}}_x) \end{bmatrix}, \tag{34}$$

$$\mathbf{\Omega}_{g,R} = \begin{bmatrix} (-\tilde{\mathbf{K}}_x\tilde{\mathbf{K}}_y) & (\tilde{\mathbf{K}}_x^2 - [\![\varepsilon_{r,g}]\!]) \\ ([\![\varepsilon_{r,g}^{-1}]\!]^{-1} - \tilde{\mathbf{K}}_y^2) & (\tilde{\mathbf{K}}_y\tilde{\mathbf{K}}_x) \end{bmatrix}, \tag{35}$$

and

$$\mathbf{\Omega}_{g,LR}^2 = \begin{bmatrix} \tilde{\mathbf{K}}_y^2 + (\tilde{\mathbf{K}}_x [\![\varepsilon_{r,g}]\!]^{-1}\tilde{\mathbf{K}}_x - \mathbf{I})[\![\varepsilon_{r,g}^{-1}]\!]^{-1} & \tilde{\mathbf{K}}_x([\![\varepsilon_{r,g}]\!]^{-1}\tilde{\mathbf{K}}_y[\![\varepsilon_{r,g}]\!] - \tilde{\mathbf{K}}_y) \\ \tilde{\mathbf{K}}_y([\![\varepsilon_{r,g}]\!]^{-1}\tilde{\mathbf{K}}_x[\![\varepsilon_{r,g}^{-1}]\!]^{-1} - \tilde{\mathbf{K}}_x) & \tilde{\mathbf{K}}_x^2 + (\tilde{\mathbf{K}}_y[\![\varepsilon_{r,g}]\!]^{-1}\tilde{\mathbf{K}}_y - \mathbf{I})[\![\varepsilon_{r,g}]\!] \end{bmatrix}, \tag{36}$$

and

$$\tilde{\mathbf{K}}_r = \begin{bmatrix} \tilde{k}_{r,(-N,-M)} & 0 & \cdots & 0 \\ 0 & \tilde{k}_{r,(-N,-M+1)} & \cdots & 0 \\ \vdots & \vdots & \ddots & \vdots \\ 0 & 0 & \cdots & \tilde{k}_{r,(N,M)} \end{bmatrix}, \quad r \in \{x, y\}, \tag{37}$$

and $[\![\ ]\!]$ is the convolution (a.k.a Toeplitz) matrix: $[\![\varepsilon_{r,g}]\!]$ and $[\![\varepsilon_{r,g}^{-1}]\!]^{-1}$ are convolution matrices composed of Fourier coefficients of permittivity and one-over-permittivity (by the inverse rule presented in (Li, 1996) and (Li, 2014)).

Equation 33 is a typical form of the eigendecomposition of a matrix. The vector $[\mathfrak{S}_{g,\vec{x}} \cdot e^{-j\tilde{k}_{g,z}\tilde{z}} \quad \mathfrak{S}_{g,\vec{y}} \cdot e^{-j\tilde{k}_{g,z}\tilde{z}}]^T$ is an eigenvector of $\mathbf{\Omega}_{g,LR}^2$ and $j\tilde{k}_{g,z}$ is the positive square root of the eigenvalues. This intuitively shows how the eigenvalues are connected to the Z-directional wavevectors.

It is also possible to use $\mathbf{S}_{g,\vec{x}}(\tilde{z})$ and $\mathbf{S}_{g,\vec{y}}(\tilde{z})$ instead of $\mathfrak{S}_{g,\vec{x}}$ and $\mathfrak{U}_{g,\vec{x}}$ because they satisfy the following relations:

$$\frac{\partial^2}{\partial(\tilde{z})^2} \begin{bmatrix} \mathbf{S}_{g,\vec{x}}(\tilde{z}) \\ \mathbf{S}_{g,\vec{y}}(\tilde{z}) \end{bmatrix} = \frac{\partial^2}{\partial(\tilde{z})^2} \begin{bmatrix} \mathfrak{S}_{g,\vec{x}} \cdot e^{-j\tilde{k}_{g,z}\tilde{z}} \\ \mathfrak{S}_{g,\vec{y}} \cdot e^{-j\tilde{k}_{g,z}\tilde{z}} \end{bmatrix} = (-j\tilde{k}_{g,z})^2 \begin{bmatrix} \mathfrak{S}_{g,\vec{x}} \cdot e^{-j\tilde{k}_{g,z}\tilde{z}} \\ \mathfrak{S}_{g,\vec{y}} \cdot e^{-j\tilde{k}_{g,z}\tilde{z}} \end{bmatrix}. \tag{38}$$

Hence it is just a matter of choice and we will use PDE form ($\mathbf{S}_g$ and $\mathbf{U}_g$) for the seamless connection to the 1D conical case in the previous work (Moharam et al., 1995a). Then Equations 31, 32 and 33 become

$$\frac{\partial}{\partial(\tilde{z})} \begin{bmatrix} \mathbf{S}_{g,\vec{x}}(\tilde{z}) \\ \mathbf{S}_{g,\vec{y}}(\tilde{z}) \end{bmatrix} = \mathbf{\Omega}_{g,L} \begin{bmatrix} \mathbf{U}_{g,\vec{x}}(\tilde{z}) \\ \mathbf{U}_{g,\vec{y}}(\tilde{z}) \end{bmatrix}, \tag{39}$$

$$\frac{\partial}{\partial(\tilde{z})} \begin{bmatrix} \mathbf{U}_{g,\vec{x}}(\tilde{z}) \\ \mathbf{U}_{g,\vec{y}}(\tilde{z}) \end{bmatrix} = \mathbf{\Omega}_{g,R} \begin{bmatrix} \mathbf{S}_{g,\vec{x}}(\tilde{z}) \\ \mathbf{S}_{g,\vec{y}}(\tilde{z}) \end{bmatrix}, \tag{40}$$

$$\frac{\partial^2}{\partial(\tilde{z})^2} \begin{bmatrix} \mathbf{S}_{g,\vec{x}}(\tilde{z}) \\ \mathbf{S}_{g,\vec{y}}(\tilde{z}) \end{bmatrix} = \mathbf{\Omega}_{g,LR}^2 \begin{bmatrix} \mathbf{S}_{g,\vec{x}}(\tilde{z}) \\ \mathbf{S}_{g,\vec{y}}(\tilde{z}) \end{bmatrix}, \tag{41}$$

where Equation (41) is the second order matrix differential equation which has the general solution of the following form

$$\begin{bmatrix} \mathbf{S}_{g,\vec{x}}(\tilde{z}) \\ \mathbf{S}_{g,\vec{y}}(\tilde{z}) \end{bmatrix} = \boldsymbol{w}_{g,1}(c_{g,1}^{+}e^{-q_{g,1}\tilde{z}} + c_{g,1}^{-}e^{+q_{g,1}\tilde{z}}) + \cdots + \boldsymbol{w}_{g,\xi}(c_{g,\xi}^{+}e^{-q_{g,\xi}\tilde{z}} + c_{g,\xi}^{-}e^{+q_{g,\xi}\tilde{z}}) \tag{42}$$

$$= \sum_{i=1}^{\xi} \boldsymbol{w}_{g,i}(c_{g,i}^{+}e^{-q_{g,i}\tilde{z}} + c_{g,i}^{-}e^{+q_{g,i}\tilde{z}}), \tag{43}$$

where $\xi = (2M + 1)(2N + 1)$, the total number of harmonics, and $\boldsymbol{w}_g$ is the eigenvector, $q_g$ is the positive square root of the corresponding eigenvalue $(j\tilde{k}_{g,z})$ and $c_g^{\pm}$ are the coefficients (amplitudes) of the mode in each propagating direction (+Z and -Z direction). This can be written in matrix form

$$\begin{bmatrix} \mathbf{S}_{g,\vec{x}}(\tilde{z}) \\ \mathbf{S}_{g,\vec{y}}(\tilde{z}) \end{bmatrix} = \mathbf{W}_g \mathbf{Q}_g^{-} \mathbf{c}_g^{+} + \mathbf{W}_g \mathbf{Q}_g^{+} \mathbf{c}_g^{-} \tag{44}$$

$$= \mathbf{W}_g \begin{bmatrix} \mathbf{Q}_g^{-} & \mathbf{Q}_g^{+} \end{bmatrix} \begin{bmatrix} \mathbf{c}_g^{+} \\ \mathbf{c}_g^{-} \end{bmatrix}, \tag{45}$$

$$= \begin{bmatrix} \mathbf{W}_{g,11} & \mathbf{W}_{g,12} \\ \mathbf{W}_{g,21} & \mathbf{W}_{g,22} \end{bmatrix} \begin{bmatrix} \mathbf{Q}_{g,1}^{-} & 0 & \mathbf{Q}_{g,1}^{+} & 0 \\ 0 & \mathbf{Q}_{g,2}^{-} & 0 & \mathbf{Q}_{g,2}^{+} \end{bmatrix} \begin{bmatrix} \mathbf{c}_{g,1}^{+} \\ \mathbf{c}_{g,2}^{+} \\ \mathbf{c}_{g,1}^{-} \\ \mathbf{c}_{g,2}^{-} \end{bmatrix}, \tag{46}$$

where $\mathbf{Q}_g^{\pm}$ are the diagonal matrices with the exponential of eigenvalues

$$\mathbf{Q}_g^{\pm} = \begin{bmatrix} e^{\pm q_{g,1}} & & 0 \\ & \ddots & \\ 0 & & e^{\pm q_{g,\xi}} \end{bmatrix}, \tag{47}$$

and $\mathbf{W}_g$ is the matrix that has the eigenvectors in columns and $\mathbf{c}_g^{\pm}$ are the vectors of the coefficients.

Now we can find the general solution of the magnetic field that shares same $\mathbf{Q}_g$ and $\mathbf{c}_g^{\pm}$ with the electric field in corresponding mode. It can be written in a similar form of Equation 44 as

$$\begin{bmatrix} \mathbf{U}_{g,\vec{x}}(\tilde{z}) \\ \mathbf{U}_{g,\vec{y}}(\tilde{z}) \end{bmatrix} = -\mathbf{V}_g \mathbf{Q}_g^{-} \mathbf{c}_g^{+} + \mathbf{V}_g \mathbf{Q}_g^{+} \mathbf{c}_g^{-}. \tag{48}$$

The negative sign in the first term was given to adjust the direction of the curl operation, $E \times H$, to be in accordance with the wave propagation direction, $\tilde{k}_{g,z}$. By substituting Equations 44 and 48 into Equation 40, we can get

$$\mathbf{V}_g = \boldsymbol{\Omega}_{g,R} \mathbf{W}_g \mathbf{q}_g^{-1}, \tag{49}$$

where $\mathbf{q}_g$ is the diagonal matrix with the eigenvalues. This can be written in matrix form

$$\mathbf{V}_g = \begin{bmatrix} \mathbf{V}_{g,11} & \mathbf{V}_{g,12} \\ \mathbf{V}_{g,21} & \mathbf{V}_{g,22} \end{bmatrix} = \begin{bmatrix} -\tilde{\mathbf{K}}_x \tilde{\mathbf{K}}_y & \tilde{\mathbf{K}}_x^2 - [\![\varepsilon_{r,g}]\!] \\ [\![\varepsilon_{r,g}^{-1}]\!]^{-1} - \tilde{\mathbf{K}}_y^2 & \tilde{\mathbf{K}}_y \tilde{\mathbf{K}}_x \end{bmatrix} \begin{bmatrix} \mathbf{W}_{g,11} & \mathbf{W}_{g,12} \\ \mathbf{W}_{g,21} & \mathbf{W}_{g,22} \end{bmatrix} \begin{bmatrix} \mathbf{q}_{g,1} & 0 \\ 0 & \mathbf{q}_{g,2} \end{bmatrix}^{-1}. \tag{50}$$

## G.4 CONNECTING LAYERS

Once the eigenmodes of each grating layer are identified, the transfer matrix method (TMM) can be utilized to determine the Rayleigh coefficients $(\mathbf{R}_s, \mathbf{R}_p, \mathbf{T}_s, \mathbf{T}_p)$ and the diffraction efficiencies. TMM effectively represents this process as a matrix multiplication, where the transfer matrix is constructed by considering the interaction between the eigenmodes of neighboring layers. This matrix accounts for the energy transfer and phase shift between the eigenmodes, and it is used to propagate the electromagnetic fields through the entire periodic structure.

From the boundary conditions, the systems of equations consisting of the in-plane (tangential) field components $(\mathbf{E}_s, \mathbf{E}_p, \mathbf{H}_s, \mathbf{H}_p)$ can be described at each layer interface. We will first consider the case of a single grating layer cladded with the superstrate and substrate, then expand to multilayer structure. At the input boundary $(z = 0)$:

$$\begin{bmatrix} \sin\psi \, \boldsymbol{\delta}_{00} \\ \cos\psi \, \cos\theta \, \boldsymbol{\delta}_{00} \\ j\sin\psi \, \mathbf{n}_I \cos\theta \, \boldsymbol{\delta}_{00} \\ -j\cos\psi \, \mathbf{n}_I \, \boldsymbol{\delta}_{00} \end{bmatrix} + \begin{bmatrix} \mathbf{I} & 0 \\ 0 & -j\mathbf{Z}_I \\ -j\mathbf{Y}_I & 0 \\ 0 & \mathbf{I} \end{bmatrix} \begin{bmatrix} \mathbf{R}_s \\ \mathbf{R}_p \end{bmatrix} = \begin{bmatrix} \mathbf{W}_{g,ss} & \mathbf{W}_{g,sp} & \mathbf{W}_{g,ss}\mathbf{X}_{g,1} & \mathbf{W}_{g,sp}\mathbf{X}_{g,2} \\ \mathbf{W}_{g,ps} & \mathbf{W}_{g,pp} & \mathbf{W}_{g,ps}\mathbf{X}_{g,1} & \mathbf{W}_{g,pp}\mathbf{X}_{g,2} \\ \mathbf{V}_{g,ss} & \mathbf{V}_{g,sp} & -\mathbf{V}_{g,ss}\mathbf{X}_{g,1} & -\mathbf{V}_{g,sp}\mathbf{X}_{g,2} \\ \mathbf{V}_{g,ps} & \mathbf{V}_{g,pp} & -\mathbf{V}_{g,ps}\mathbf{X}_{g,1} & -\mathbf{V}_{g,pp}\mathbf{X}_{g,2} \end{bmatrix} \begin{bmatrix} \mathbf{c}_{g,1}^{+} \\ \mathbf{c}_{g,2}^{+} \\ \mathbf{c}_{g,1}^{-} \\ \mathbf{c}_{g,2}^{-} \end{bmatrix},$$

$$\tag{51}$$

and at the output boundary ($z = d$):

$$
\begin{bmatrix}
\mathbf{W}_{g,ss}\mathbf{X}_{g,1} & \mathbf{W}_{g,sp}\mathbf{X}_{g,2} & \mathbf{W}_{g,ss} & \mathbf{W}_{g,sp} \\
\mathbf{W}_{g,ps}\mathbf{X}_{g,1} & \mathbf{W}_{g,pp}\mathbf{X}_{g,2} & \mathbf{W}_{g,ps} & \mathbf{W}_{g,pp} \\
\mathbf{V}_{g,ss}\mathbf{X}_{g,1} & \mathbf{V}_{g,sp}\mathbf{X}_{g,2} & -\mathbf{V}_{g,ss} & -\mathbf{V}_{g,sp} \\
\mathbf{V}_{g,ps}\mathbf{X}_{g,1} & \mathbf{V}_{g,pp}\mathbf{X}_{g,2} & -\mathbf{V}_{g,ps} & -\mathbf{V}_{g,pp}
\end{bmatrix}
\begin{bmatrix}
\mathbf{c}_{g,1}^{+} \\
\mathbf{c}_{g,2}^{+} \\
\mathbf{c}_{g,1}^{-} \\
\mathbf{c}_{g,2}^{-}
\end{bmatrix}
=
\begin{bmatrix}
\mathbf{I} & \mathbf{0} \\
\mathbf{0} & j\mathbf{Z}_{\mathrm{II}} \\
j\mathbf{Y}_{\mathrm{II}} & \mathbf{0} \\
\mathbf{0} & \mathbf{I}
\end{bmatrix}
\begin{bmatrix}
\mathbf{T}_s \\
\mathbf{T}_p
\end{bmatrix},
\tag{52}
$$

where $\boldsymbol{\delta}_{00}$ is the Kronecker delta function that has 1 at the $(0,0)^{th}$ order and 0 elsewhere.

Here, the variables used above are defined: $\mathbf{X}_{g,1}, \mathbf{X}_{g,2}$ are the diagonal matrices

$$
\mathbf{X}_{g,1} =
\begin{bmatrix}
e^{-k_0 q_{g,1,1} d_g} & & 0 \\
& \ddots & \\
0 & & e^{-k_0 q_{g,1,\xi} d_g}
\end{bmatrix}, \quad
\mathbf{X}_{g,2} =
\begin{bmatrix}
e^{-k_0 q_{g,2,1} d_g} & & 0 \\
& \ddots & \\
0 & & e^{-k_0 q_{g,2,\xi} d_g}
\end{bmatrix},
\tag{53}
$$

where $d_g$ is the thickness of the grating layer, and $\mathbf{Y}_{\mathrm{I}}$ and $\mathbf{Z}_{\mathrm{I}}$ are

$$
\mathbf{Y}_{\mathrm{I}} =
\begin{bmatrix}
\tilde{k}_{\mathrm{I},z,(-N,-M)} & & 0 \\
& \ddots & \\
0 & & \tilde{k}_{\mathrm{I},z,(N,M)}
\end{bmatrix}, \quad
\mathbf{Z}_{\mathrm{I}} = \frac{1}{(\mathrm{n}_{\mathrm{I}})^2}
\begin{bmatrix}
\tilde{k}_{\mathrm{I},z,(-N,-M)} & & 0 \\
& \ddots & \\
0 & & \tilde{k}_{\mathrm{I},z,(N,M)}
\end{bmatrix},
\tag{54}
$$

and $\mathbf{Y}_{\mathrm{II}}$ and $\mathbf{Z}_{\mathrm{II}}$ are

$$
\mathbf{Y}_{\mathrm{II}} =
\begin{bmatrix}
\tilde{k}_{\mathrm{II},z,(-N,-M)} & & 0 \\
& \ddots & \\
0 & & \tilde{k}_{\mathrm{II},z,(N,M)}
\end{bmatrix}, \quad
\mathbf{Z}_{\mathrm{II}} = \frac{1}{(\mathrm{n}_{\mathrm{II}})^2}
\begin{bmatrix}
\tilde{k}_{\mathrm{II},z,(-N,-M)} & & 0 \\
& \ddots & \\
0 & & \tilde{k}_{\mathrm{II},z,(N,M)}
\end{bmatrix}.
\tag{55}
$$

Here, new set of $\mathbf{W}_g$ and $\mathbf{V}_g$ on SP basis $\{\hat{s}, \hat{p}\}$ are introduced which are recombined from the set of $\mathbf{W}_g$ and $\mathbf{V}_g$ from XY basis $\{\hat{x}, \hat{y}\}$:

$$
\mathbf{W}_{g,ss} = \mathbf{F}_c \mathbf{W}_{g,21} - \mathbf{F}_s \mathbf{W}_{g,11}, \qquad \mathbf{W}_{g,sp} = \mathbf{F}_c \mathbf{W}_{g,22} - \mathbf{F}_s \mathbf{W}_{g,12},
\tag{56}
$$

$$
\mathbf{W}_{g,ps} = \mathbf{F}_c \mathbf{W}_{g,11} + \mathbf{F}_s \mathbf{W}_{g,21}, \qquad \mathbf{W}_{g,pp} = \mathbf{F}_c \mathbf{W}_{g,12} + \mathbf{F}_s \mathbf{W}_{g,22},
\tag{57}
$$

$$
\mathbf{V}_{g,ss} = \mathbf{F}_c \mathbf{V}_{g,11} + \mathbf{F}_s \mathbf{V}_{g,21}, \qquad \mathbf{V}_{g,sp} = \mathbf{F}_c \mathbf{V}_{g,12} + \mathbf{F}_s \mathbf{V}_{g,22},
\tag{58}
$$

$$
\mathbf{V}_{g,ps} = \mathbf{F}_c \mathbf{V}_{g,21} - \mathbf{F}_s \mathbf{V}_{g,11}, \qquad \mathbf{V}_{g,pp} = \mathbf{F}_c \mathbf{V}_{g,22} - \mathbf{F}_s \mathbf{V}_{g,12},
\tag{59}
$$

with $\mathbf{F}_c$ and $\mathbf{F}_s$ being diagonal matrices with the diagonal elements $\cos\varphi_{(n,m)}$ and $\sin\varphi_{(n,m)}$, respectively, where

$$
\varphi_{(n,m)} = \tan^{-1}(k_{y,n}/k_{x,m}).
\tag{60}
$$

Equations 51 and 52 can be reduced to one set of equations by eliminating $\mathbf{c}_{1,2}^{\pm}$:

$$
\begin{bmatrix}
\sin\psi\, \boldsymbol{\delta}_{00} \\
\cos\psi\, \cos\theta\, \boldsymbol{\delta}_{00} \\
j\sin\psi\, \mathrm{n}_{\mathrm{I}}\cos\theta\, \boldsymbol{\delta}_{00} \\
-j\cos\psi\, \mathrm{n}_{\mathrm{I}}\, \boldsymbol{\delta}_{00}
\end{bmatrix}
+
\begin{bmatrix}
\mathbf{I} & \mathbf{0} \\
\mathbf{0} & -j\mathbf{Z}_I \\
-j\mathbf{Y}_I & \mathbf{0} \\
\mathbf{0} & \mathbf{I}
\end{bmatrix}
\begin{bmatrix}
\mathbf{R}_s \\
\mathbf{R}_p
\end{bmatrix}
=
\begin{bmatrix}
\mathbb{W} & \mathbb{W}\mathbb{X} \\
\mathbb{V} & -\mathbb{V}\mathbb{X}
\end{bmatrix}
\begin{bmatrix}
\mathbb{W}\mathbb{X} & \mathbb{W} \\
\mathbb{V}\mathbb{X} & -\mathbb{V}
\end{bmatrix}^{-1}
\begin{bmatrix}
\mathbb{F} \\
\mathbb{G}
\end{bmatrix}
\begin{bmatrix}
\mathbf{T}_s \\
\mathbf{T}_p
\end{bmatrix},
\tag{61}
$$

where

$$
\mathbb{W} =
\begin{bmatrix}
\mathbf{W}_{g,ss} & \mathbf{W}_{g,sp} \\
\mathbf{W}_{g,ps} & \mathbf{W}_{g,pp}
\end{bmatrix}, \quad
\mathbb{V} =
\begin{bmatrix}
\mathbf{V}_{g,ss} & \mathbf{V}_{g,sp} \\
\mathbf{V}_{g,ps} & \mathbf{V}_{g,pp}
\end{bmatrix}, \quad
\mathbb{X} =
\begin{bmatrix}
\mathbf{X}_{g,1} & \mathbf{0} \\
\mathbf{0} & \mathbf{X}_{g,2}
\end{bmatrix}, \quad
\mathbb{F} =
\begin{bmatrix}
\mathbf{I} & \mathbf{0} \\
\mathbf{0} & j\mathbf{Z}_{\mathrm{II}}
\end{bmatrix}, \quad
\mathbb{G} =
\begin{bmatrix}
j\mathbf{Y}_{\mathrm{II}} & \mathbf{0} \\
\mathbf{0} & \mathbf{I}
\end{bmatrix}.
\tag{62}
$$

This equation for a single layer grating can be simply extended to a multi-layer system as the following:

$$
\begin{bmatrix}
\sin\psi\, \boldsymbol{\delta}_{00} \\
\cos\psi\, \cos\theta\, \boldsymbol{\delta}_{00} \\
j\sin\psi\, \mathrm{n}_{\mathrm{I}}\cos\theta\, \boldsymbol{\delta}_{00} \\
-j\cos\psi\, \mathrm{n}_{\mathrm{I}}\, \boldsymbol{\delta}_{00}
\end{bmatrix}
+
\begin{bmatrix}
\mathbf{I} & \mathbf{0} \\
\mathbf{0} & -j\mathbf{Z}_I \\
-j\mathbf{Y}_I & \mathbf{0} \\
\mathbf{0} & \mathbf{I}
\end{bmatrix}
\begin{bmatrix}
\mathbf{R}_s \\
\mathbf{R}_p
\end{bmatrix}
=
\prod_{\ell=1}^{L}
\begin{bmatrix}
\mathbb{W}_\ell & \mathbb{W}_\ell\mathbb{X}_\ell \\
\mathbb{V}_\ell & -\mathbb{V}_\ell\mathbb{X}_\ell
\end{bmatrix}
\begin{bmatrix}
\mathbb{W}_\ell\mathbb{X}_\ell & \mathbb{W}_\ell \\
\mathbb{V}_\ell\mathbb{X}_\ell & -\mathbb{V}_\ell
\end{bmatrix}^{-1}
\begin{bmatrix}
\mathbb{F}_{L+1} \\
\mathbb{G}_{L+1}
\end{bmatrix}
\begin{bmatrix}
\mathbf{T}_s \\
\mathbf{T}_p
\end{bmatrix},
\tag{63}
$$

where $L$ is the number of layers and

$$
\mathbb{F}_{L+1} =
\begin{bmatrix}
\mathbf{I} & \mathbf{0} \\
\mathbf{0} & j\mathbf{Z}_{\mathrm{II}}
\end{bmatrix}, \quad
\mathbb{G}_{L+1} =
\begin{bmatrix}
j\mathbf{Y}_{\mathrm{II}} & \mathbf{0} \\
\mathbf{0} & \mathbf{I}
\end{bmatrix}.
\tag{64}
$$

Since we have four matrix equations for four unknown coefficients ($\mathbf{R}_s$, $\mathbf{R}_p$, $\mathbf{T}_s$, $\mathbf{T}_p$), they can be derived and used for calculating diffraction efficiencies (also called the reflectance and transmittance).

The diffraction efficiency is the ratio of the power flux in propagating direction between incidence and diffracted wave of interest. It can be calculated by time-averaged Poynting vector (Liu & Fan, 2012; Hugonin & Lalanne, 2021; Rumpf, 2006):

$$P = \frac{1}{2} \operatorname{Re}(E \times H^*), \tag{65}$$

where $^*$ is the complex conjugate. Now we can find the total power of the incident wave as a sum of the power of TE wave and TM wave:

$$
\begin{aligned}
P^{inc} &= P_s^{inc} + P_p^{inc} \\
&= \frac{1}{2} \operatorname{Re}\left[(E_s \times H_s^*) + (E_p \times H_p^*)\right] \\
&= \frac{1}{2} \operatorname{Re}\left[(\sin\psi \cdot \sin\psi \, \mathbf{n}_\mathrm{I} \, \cos\theta) + (\cos\psi \, \cos\theta \cdot \cos\psi \, \mathbf{n}_\mathrm{I})\right] \\
&= \frac{1}{2} \operatorname{Re}\left[(\sin^2\psi \, \mathbf{n}_\mathrm{I} \cos\theta) + (\cos^2\psi \, \mathbf{n}_\mathrm{I} \cos\theta)\right] \\
&= \frac{1}{2} \operatorname{Re}\left[(\mathbf{n}_\mathrm{I} \cos\theta)\right].
\end{aligned} \tag{66}
$$

The power in each reflected diffraction mode is

$$
\begin{aligned}
P_{n,m}^r &= P_{nm,s}^r + P_{nm,p}^r \\
&= \frac{1}{2} \operatorname{Re}\left[(E_{nm,s}^r \times (H_{nm,s}^r)^*) + (E_{nm,p}^r \times (H_{nm,p}^r)^*)\right] \\
&= \frac{1}{2} \operatorname{Re}\left[R_{nm,s} \cdot \frac{k_{\mathrm{I},z,(n,m)}}{k_0} R_{nm,s}^* + \frac{k_{\mathrm{I},z,(n,m)}}{k_0 \mathbf{n}_\mathrm{I}^2} R_{nm,p} \cdot R_{nm,p}^*\right] \\
&= \frac{1}{2} \operatorname{Re}\left[R_{nm,s} R_{nm,s}^* \cdot \frac{k_{\mathrm{I},z,(n,m)}}{k_0} + R_{nm,p} R_{nm,p}^* \cdot \frac{k_{\mathrm{I},z,(n,m)}}{k_0 \mathbf{n}_\mathrm{I}^2}\right],
\end{aligned} \tag{67}
$$

and the power in each transmitted diffraction mode is

$$
\begin{aligned}
P_{n,m}^t &= P_{nm,s}^t + P_{nm,p}^t \\
&= \frac{1}{2} \operatorname{Re}\left[(E_{nm,s}^t \times (H_{nm,s}^t)^*) + (E_{nm,p}^t \times (H_{nm,p}^t)^*)\right] \\
&= \frac{1}{2} \operatorname{Re}\left[T_{nm,s} \cdot \frac{k_{\mathrm{II},z,(n,m)}}{k_0} T_{nm,s}^* + \frac{k_{\mathrm{II},z,(n,m)}}{k_0 \mathbf{n}_\mathrm{II}^2} T_{nm,p} \cdot T_{nm,p}^*\right] \\
&= \frac{1}{2} \operatorname{Re}\left[T_{nm,s} T_{nm,s}^* \cdot \frac{k_{\mathrm{II},z,(n,m)}}{k_0} + T_{nm,p} T_{nm,p}^* \cdot \frac{k_{\mathrm{II},z,(n,m)}}{k_0 \mathbf{n}_\mathrm{II}^2}\right].
\end{aligned} \tag{68}
$$

Since the diffraction efficiency is the ratio between them ($P_{out}/P_{inc}$), we can get the efficiencies of reflected and transmitted waves:

$$DE_{r,(n,m)} = |R_{s,(n,m)}|^2 \operatorname{Re}\left(\frac{k_{\mathrm{I},z,(n,m)}}{k_0 \mathbf{n}_\mathrm{I} \cos\theta}\right) + |R_{p,(n,m)}|^2 \operatorname{Re}\left(\frac{k_{\mathrm{I},z,(n,m)}/\mathbf{n}_\mathrm{I}^2}{k_0 \mathbf{n}_\mathrm{I} \cos\theta}\right), \tag{69}$$

$$DE_{t,(n,m)} = |T_{s,(n,m)}|^2 \operatorname{Re}\left(\frac{k_{\mathrm{II},z,(n,m)}}{k_0 \mathbf{n}_\mathrm{I} \cos\theta}\right) + |T_{p,(n,m)}|^2 \operatorname{Re}\left(\frac{k_{\mathrm{II},z,(n,m)}/\mathbf{n}_\mathrm{II}^2}{k_0 \mathbf{n}_\mathrm{I} \cos\theta}\right). \tag{70}$$

## G.5 ENHANCED TRANSMITTANCE MATRIX METHOD

As addressed in (Moharam et al., 1995b; Li, 1993; Popov & Nevière, 2000), solving Equation 63 may suffer from the numerical instability coming from the inversion of almost singular matrix when $\mathbb{X}_\ell$ has a very small and

possibly numerically zero value. `meent` adopted Enhanced Transmittance Matrix Method (ETM) (Moharam et al., 1995b) to overcome this by avoiding the inversion of $\mathbb{X}_\ell$.

The technique is sequentially applied from the last layer to the first layer. In Equation 63, the set of modes at the bottom interface of the last layer ($\ell = L$) is

$$
\begin{bmatrix} \mathbb{W}_L & \mathbb{W}_L \mathbb{X}_L \\ \mathbb{V}_L & -\mathbb{V}_L \mathbb{X}_L \end{bmatrix} \begin{bmatrix} \mathbb{W}_L \mathbb{X}_L & \mathbb{W}_L \\ \mathbb{V}_L \mathbb{X}_L & -\mathbb{V}_L \end{bmatrix}^{-1} \begin{bmatrix} \mathbb{F}_{L+1} \\ \mathbb{G}_{L+1} \end{bmatrix} \begin{bmatrix} \mathbf{T}_s \\ \mathbf{T}_p \end{bmatrix}
$$
$$
= \begin{bmatrix} \mathbb{W}_L & \mathbb{W}_L \mathbb{X}_L \\ \mathbb{V}_L & -\mathbb{V}_L \mathbb{X}_L \end{bmatrix} \begin{bmatrix} \mathbb{X}_L^{-1} & \mathbb{0} \\ \mathbb{0} & \mathbb{I} \end{bmatrix} \begin{bmatrix} \mathbb{W}_L & \mathbb{W}_L \\ \mathbb{V}_L & -\mathbb{V}_L \end{bmatrix}^{-1} \begin{bmatrix} \mathbb{F}_{L+1} \\ \mathbb{G}_{L+1} \end{bmatrix} \begin{bmatrix} \mathbf{T}_s \\ \mathbf{T}_p \end{bmatrix}.
$$
(71)

The matrix to be inverted can be decomposed into two matrices by isolating $\mathbb{X}_L$, which is the potential source of the numerical instability. The right-hand side can be shortened with new variables $\mathbb{A}_L, \mathbb{B}_L$:

$$
\begin{bmatrix} \mathbb{A}_L \\ \mathbb{B}_L \end{bmatrix} = \begin{bmatrix} \mathbb{W}_L & \mathbb{W}_{\mathbb{L}} \\ \mathbb{V}_{\mathbb{L}} & -\mathbb{V}_L \end{bmatrix}^{-1} \begin{bmatrix} \mathbb{F}_{L+1} \\ \mathbb{G}_{L+1} \end{bmatrix},
$$
(72)

then the right-hand side of Equation 71 becomes

$$
\begin{bmatrix} \mathbb{W}_L & \mathbb{W}_L \mathbb{X}_L \\ \mathbb{V}_L & -\mathbb{V}_L \mathbb{X}_L \end{bmatrix} \begin{bmatrix} \mathbb{X}_L^{-1} & \mathbb{0} \\ \mathbb{0} & \mathbb{I} \end{bmatrix} \begin{bmatrix} \mathbb{A}_L \\ \mathbb{B}_L \end{bmatrix} \begin{bmatrix} \mathbf{T}_s \\ \mathbf{T}_p \end{bmatrix}.
$$
(73)

We can avoid the inversion of $\mathbb{X}_L$ by introducing the substitution $\mathbf{T}_s = \mathbb{A}_L^{-1}\mathbb{X}_L\mathbf{T}_{s,L}$ and $\mathbf{T}_p = \mathbb{A}_L^{-1}\mathbb{X}_L\mathbf{T}_{p,L}$. Equation 73 then becomes

$$
\begin{bmatrix} \mathbb{W}_L & \mathbb{W}_L \mathbb{X}_L \\ \mathbb{V}_L & -\mathbb{V}_L \mathbb{X}_L \end{bmatrix} \begin{bmatrix} \mathbb{X}_L^{-1} & \mathbb{0} \\ \mathbb{0} & \mathbb{I} \end{bmatrix} \begin{bmatrix} \mathbb{A}_L \\ \mathbb{B}_L \end{bmatrix} \mathbb{A}_L^{-1}\mathbb{X}_L \begin{bmatrix} \mathbf{T}_{s,L} \\ \mathbf{T}_{p,L} \end{bmatrix}
$$
$$
= \begin{bmatrix} \mathbb{W}_L & \mathbb{W}_L \mathbb{X}_L \\ \mathbb{V}_L & -\mathbb{V}_L \mathbb{X}_L \end{bmatrix} \begin{bmatrix} \mathbb{X}_L^{-1} & \mathbb{0} \\ \mathbb{0} & \mathbb{I} \end{bmatrix} \begin{bmatrix} \mathbb{X}_L \\ \mathbb{B}_L \mathbb{A}_L^{-1}\mathbb{X}_L \end{bmatrix} \begin{bmatrix} \mathbf{T}_{s,L} \\ \mathbf{T}_{p,L} \end{bmatrix}
$$
$$
= \begin{bmatrix} \mathbb{W}_L & \mathbb{W}_L \mathbb{X}_L \\ \mathbb{V}_L & -\mathbb{V}_L \mathbb{X}_L \end{bmatrix} \begin{bmatrix} \mathbb{I} \\ \mathbb{B}_L \mathbb{A}_L^{-1}\mathbb{X}_L \end{bmatrix} \begin{bmatrix} \mathbf{T}_{s,L} \\ \mathbf{T}_{p,L} \end{bmatrix}
$$
$$
= \begin{bmatrix} \mathbb{W}_L (\mathbb{I} + \mathbb{X}_L \mathbb{B}_L \mathbb{A}_L^{-1}\mathbb{X}) \\ \mathbb{V}_L (\mathbb{I} - \mathbb{X}_L \mathbb{B}_L \mathbb{A}_L^{-1}\mathbb{X}) \end{bmatrix} \begin{bmatrix} \mathbf{T}_{s,L} \\ \mathbf{T}_{p,L} \end{bmatrix}
$$
$$
= \begin{bmatrix} \mathbb{F}_L \\ \mathbb{G}_L \end{bmatrix} \begin{bmatrix} \mathbf{T}_{s,L} \\ \mathbf{T}_{p,L} \end{bmatrix}.
$$
(74)

These steps can be repeated until the iteration gets to the first layer ($\ell = 1$), then the form becomes

$$
\begin{bmatrix} \sin\psi\,\boldsymbol{\delta}_{00} \\ \cos\psi\,\cos\theta\,\boldsymbol{\delta}_{00} \\ j\sin\psi\,n_{\mathrm{I}}\,\cos\theta\,\boldsymbol{\delta}_{00} \\ -j\cos\psi\,n_{\mathrm{I}}\,\boldsymbol{\delta}_{00} \end{bmatrix} + \begin{bmatrix} \mathbf{I} & \mathbf{0} \\ \mathbf{0} & -j\mathbf{Z}_I \\ -j\mathbf{Y}_I & \mathbf{0} \\ \mathbf{0} & \mathbf{I} \end{bmatrix} \begin{bmatrix} \mathbf{R}_s \\ \mathbf{R}_p \end{bmatrix} = \begin{bmatrix} \mathbb{F}_1 \\ \mathbb{G}_1 \end{bmatrix} \begin{bmatrix} \mathbf{T}_{s,1} \\ \mathbf{T}_{p,1} \end{bmatrix},
$$
(75)

where

$$
\begin{bmatrix} \mathbf{T}_s \\ \mathbf{T}_p \end{bmatrix} = \mathbb{A}_L^{-1}\mathbb{X}_L \cdots \mathbb{A}_\ell^{-1}\mathbb{X}_\ell \cdots \mathbb{A}_1^{-1}\mathbb{X}_1 \begin{bmatrix} \mathbf{T}_{s,1} \\ \mathbf{T}_{p,1} \end{bmatrix}.
$$

## G.6 TOPOLOGICAL DERIVATIVE VS SHAPE DERIVATIVE

AD enables the calculation of the gradient of the figure of merit (FoM) with respect to the design parameters of the device. AD in `meent` can handle both modeling type - raster and vector - with two different forms: topological derivative and shape derivative. If the raster modeling is utilized to obtain the geometry of the device, the gradients with respect to the refractive index of every pixel can be obtained through AD. This type of AD is known as the topological derivative, as the device design is updated pixel-wise and the topology is not conserved (Figure 17a). On the contrary, a shape derivative is effective for vector modeling; the FoM derivative with respect to input dimensions is obtained as depicted in Figure 17b. The shape derivative is expected to be useful for cases where the device topology is known, but dimensions of specific structures, such as the radius of a cylinder in a layer or width and length of a cuboid, are to be found by optimization.

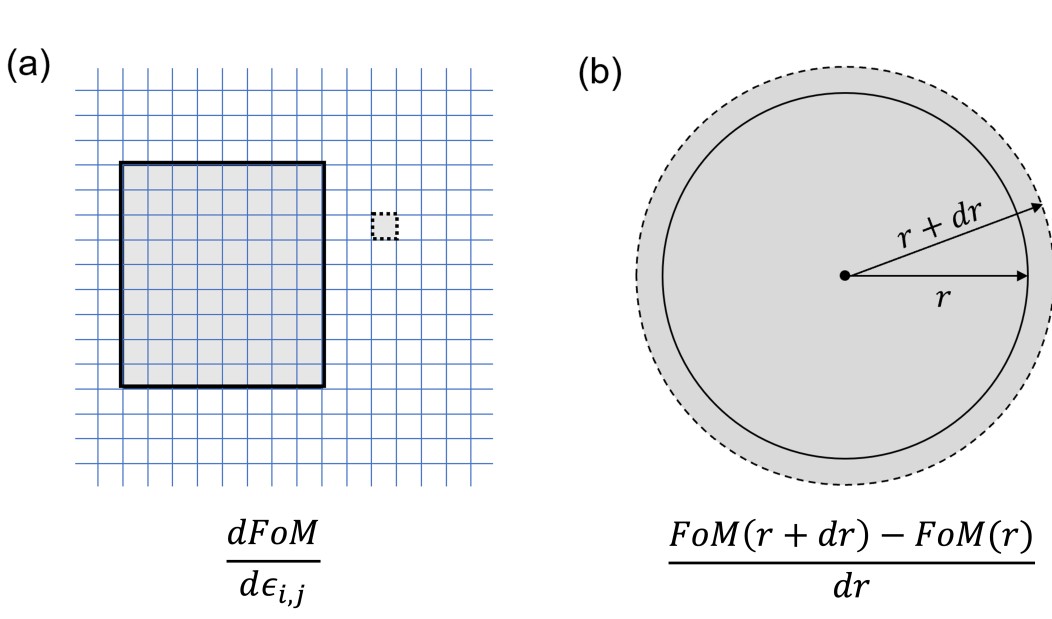

(a)

$$\frac{dFoM}{d\epsilon_{i,j}}$$

(b)

$$\frac{FoM(r+dr) - FoM(r)}{dr}$$

Figure 17: **Topological and shape derivatives.** A schematic diagram showing the difference between the (a) topological derivative and (b) shape derivative. The topological derivative results in the FoM derivative with respect to the permittivity changes of every cells in the grid and the shape derivative provides the derivative with respect to the deformations of a shape.

# H PROGRAM SEQUENCE

In this section, we will provide a detailed explanation of the functions in `meent`[4] and discuss the simulation program sequence with examples.

## H.1 INITIALIZATION

A simple way to use `meent` is using 'call_mee()' function which returns an instance of Python class that includes all the functionalities of `meent`. Simulation conditions can be set by passing parameters as arguements (args) or keyword arguements (kwargs) in this function. It is also possible to change conditions after calling instance by directly assigning desired value to the property of the instance.

```
1    # method 1: thickness setting in instance call
2    mee = meent.call_mee(backend=backend, thickness=thickness, ...)
3
4    # method 2: direct assignment
5    mee = meent.call_mee(backend=backend, ...)
6    mee.thickness = thickness
```

Code 3: Methods to set simulation conditions

Here are the descriptions of the input parameters in `meent` class:

***backend : integer***
> `meent` supports three backends: NumPy, JAX, and PyTorch.
>> - 0: NumPy (RCWA only; AD is not supported).
>> - 1: JAX.
>> - 2: PyTorch.

***grating_type : integer***
> This parameter defines the simulation space.
>> - 0: 1D grating without conical incidence ($\phi = 0$).
>> - 1: 1D grating with conical incidence.
>> - 2: 2D grating.

***pol : integer or float***
> This parameter controls the linear polarization state of the incident wave by this definition: $\psi = \pi/2 * (1 - pol)$. It can take values between 0 and 1. 0 represents fully transverse electric (TE) polarization, and 1 represents fully transverse magnetic (TM) polarization. Support for other polarization states such as the circular polarization state which involves the phase difference between TE and TM polarization will be added in the future updates.

***n_I : float***
> The refractive index of the superstrate.

***n_II : float***
> The refractive index of the substrate.

***theta : float***
> The angle of the incidence in radians.

***phi : float***
> The angle of rotation (or azimuth angle) in radians.

***wavelength : float***
> The wavelength of the incident light in vacuum. Future versions may support complex type wavelength.

***fourier_order : integer or list of integers***
> Fourier truncation order (FTO). This represents the number of Fourier harmonics in use. If *fourier_order* = $N$, this is for 1D grating and `meent` utilizes $(2N + 1)$ harmonics spanning from $-N$ to $N$:$-N, -(N-1), ..., N$. For 2D gratings, it takes a sequence $[M, N]$ as an input, where $M$ and $N$ become FTO in $X$ and $Y$ directions, respectively. Note that 1D grating can also be simulated in 2D grating system by setting $N$ as 0.

---

[4]for version 0.9.x

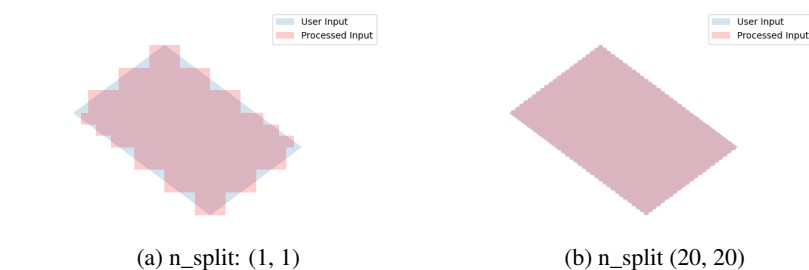

(a) n_split: (1, 1)            (b) n_split (20, 20)

Figure 18: **Rotated rectangles with approximation.** Light blue is the ideal one and light red is approximated one.

*period : list of floats*
> The period of a unit cell. For 1D grating, it is a sequence with one element which is a period in X-direction. For 2D gratings, it takes a sequence [period in $X$, period in $Y$] as an input.

*type_complex : integer*
> The datatype used in the simulation.
> - 0: complex128 (64 bit).
> - 1: complex64 (32 bit).

*device : integer*
> The selection of the device for the calculations: currently CPU and GPU are supported. At the time of writing this paper, the eigendecomposition, which is the most expensive step as $\mathcal{O}(M^3N^3)$ where $M$ and $N$ are FTO, is available only on CPU. This means GPU may not as powerful as we expect as in deep learning regime.
> - 0: CPU.
> - 1: GPU.

*fft_type : integer*
> This variable selects the type of Fourier series implementation. 0 and 1 are options for raster modeling and 2 is for vector modeling. 0 uses discrete Fourier series (DFS) while 1 and 2 use continuous Fourier series (CFS). Note that the name 'fft_type' may change since it is not correct expression.
> - 0: DFS for the raster modeling (pixel-based geometry). *fft_type* supports *improve_dft* option, which is True by default, that can prevent aliasing by increasing sampling frequency, and drives the result to approach to the result of CFS.
> - 1: CFS for the raster modeling (pixel-based geometry). This doesn't support backpropagation. Use this option for debugging or in RCWA-only situation.
> - 2: CFS for the vector modeling (object-based geometry).

*thickness : list of floats*
> The sequence of the thickness of each layer from top to bottom.

*ucell : array of {floats, complex numbers}, shape is (i, j, k)*
> The input for the raster modeling. It takes a 3D array in $(Z,Y,X)$ order, where $Z$ represents the direction of the layer stacking. In case of 1D grating, j is 1 (e.g., shape = (3,1,10) for a stack composed of 3 layers that are 1D grating).

## H.2 STRUCTURE DESIGN

meent provides two types of structure design methods: the vector modeling and the raster modeling.

### H.2.1 VECTOR MODELING

Figure 18 shows rotated rectangles drawn on XY plane. meent decomposes the geometrical figures into the collection of sub-rectangles which of each side lies on the direction of either $\hat{x}$ or $\hat{y}$. Then CFS with the sinc function is used to find the Fourier coefficients. The degree of approximation can be determined by 'n_split' option in Code 4.

To add primitives to the simulation space, users can utilize 'rectangle()' or 'rectangle_rotation()' functions which allows the insertion of desired geometry. The 'draw()' function is then employed to create the final structure, taking into account any potential overlaps between the geometries. Code 4 is the example creating a layer that has rotated rectangle.

```
1   thickness = [300.]
2   length_x = 100
3   length_y = 300
4   center = [300, 500]
5   n_index_1 = 3.48
6   n_index_2 = 1
7   base_n_index_of_layer = n_index_2
8   angle = 35 * torch.pi / 180
9   n_split = [5, 5]  # degree of approximation
10
11  length_x = torch.tensor(length_x, dtype=torch.float64, requires_grad=
    True)
12  length_y = torch.tensor(length_y, dtype=torch.float64, requires_grad=
    True)
13  thickness = torch.tensor(thickness, requires_grad=True)
14  angle = torch.tensor(angle, requires_grad=True)
15
16  obj_list = mee.rectangle_rotate(*center, length_x, length_y, *n_split
    , n_index_1, angle)
17  layer_info_list = [[base_n_index_of_layer, obj_list]]
18  mee.draw(layer_info_list)
```

Code 4: vector modeling

(a) Red rectangle comes first, blue rectangle last

(b) Blue rectangle comes first, red rectangle last

Figure 19: **The overlap of 2 rectangles in vector modeling.** The hierarchy is determined by the index of the objects in the list.

```
1    red_rect = mee.rectangle_rotate(*[400, 500], 400, 600, 20, 20, 3.5,
     0)
2    blue_rect = mee.rectangle_rotate(*[600, 500], 100, 600, 40, 40, 10,
     -20)
3
4    layer_info_list = [[2.4, red_rect + blue_rect]]   # red bottom, blue
     top
5    layer_info_list = [[2.4, blue_rect + red_rect]]   # blue bottom, red
     top
6
7    mee.draw(layer_info_list)
8    de_ri, de_ti = mee.conv_solve()
```

Code 5: overlap

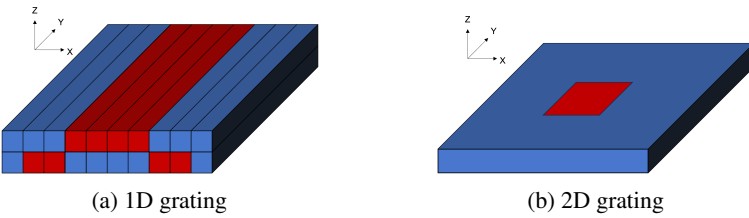

(a) 1D grating                    (b) 2D grating

Figure 20: **Raster-type structure examples.** (a) 2 layers in 1D and (b) 1 layer in 2D grating.

Code 5 and Figure 19 show how `meent` can handle the overlap of the shapes. Figure 19a and 19b have the same set of rectangles (red and blue) but they are placed in different order and this can be controlled by the function 'layer_info_list' in Code 5. It is the list that contains the base refractive index of the layer and the primitive shapes to be placed on the layer. In case of Figure 19a, red rectangle comes first in the list and blue does for Figure 19b.

### H.2.2   RASTER MODELING

We have 2 example structures of raster modeling as shown in Figure 20 and Code 6. Figure 20a is a stack of 2 layers which has 1D grating. Note that 1D grating unit cell can be defined by setting the length of the second axis to 1 as (a) in Code 6. Figure 20b is a stack of single 2D grating layer.

### H.3   ELECTROMAGNETIC SIMULATION

Electromagnetic simulation (EM simulation) in `meent` can be divided into 3 main subcategories: convolution matrix generation, Maxwell's equations computation and field calculation. The method 'conv_solve()' does both convolution matrix generation and Maxwell's equations computation sequentially. 'conv_solve_field()' method does the same and additionally calculates the field distribution of the structure. Code 7 is the example showing how to use those; 'conv_solve()' method returns the reflected and transmitted diffraction efficiencies and 'conv_solve_field()' does both and field distribution.

### H.3.1   CONVOLUTION MATRIX GENERATION

The functions for convolution matrix generation are located in 'convolution_matrix.py' file for each backend. This part transforms the structure from the real space to the Fourier space and returns a convolution matrix (also called Toeplitz matrix) of the Fourier coefficients to apply convolution operation with the E and H fields. Figure 21 shows the Fourier coefficients matrix and convolution matrix made from the coefficient matrix. Code 8 is the definition of 'conv_solve()' method and shows how the convolution matrix generation is integrated inside. As shown in the code, `meent` offers 3 different methods to get convolution matrix since each method has different input type and implementation. This can be chosen by the argument 'fft_type': 0 is for raster modeling with DFS, 1 for raster with CFS and 2 for vector with CFS.

```
1   # (a): 1D grating with 2 layers
2   ucell = np.array(
3       [
4           [[1, 1, 1, 3.48, 3.48, 3.48, 3.48, 1, 1, 1]],
5           [[1, 3.48, 3.48, 1, 1, 1, 1, 3.48, 3.48, 1]],
6       ])   # array shape: (2, 1, 10)
7
8   # (b): 2D grating with 1 layers
9   ucell = np.array(
10      [[
11          [1, 1, 1, 1, 1, 1, 1, 1, 1, 1],
12          [1, 1, 1, 3.48, 3.48, 3.48, 3.48, 1, 1, 1],
13          [1, 1, 1, 3.48, 3.48, 3.48, 3.48, 1, 1, 1],
14          [1, 1, 1, 3.48, 3.48, 3.48, 3.48, 1, 1, 1],
15          [1, 1, 1, 3.48, 3.48, 3.48, 3.48, 1, 1, 1],
16          [1, 1, 1, 3.48, 3.48, 3.48, 3.48, 1, 1, 1],
17          [1, 1, 1, 1, 1, 1, 1, 1, 1, 1],
18          [1, 1, 1, 1, 1, 1, 1, 1, 1, 1],
19      ]])   # array shape: (1, 8, 10)
20
21  mee = meent.call_mee(backend=backend, ucell=ucell)
```

Code 6: Raster modeling

```
1       mee = call_mee(backend, ...)
2
3       # generates convolution matrix and solves Maxwell's equation.
4       de_ri, de_ti = mee.conv_solve()
5
6       # generates convolution matrix, solves Maxwell's equation and
7       # reconstructs field distribution.
8       de_ri, de_ti, field_cell = mee.conv_solve_field()
```

Code 7: Method call for EM simulation

(a) Coefficients matrix      (b) Convolution matrix

Figure 21: **Material property in Fourier space.** (a) Coefficients matrix of Fourier analysis and (b) convolution matrix generated by re-arranging (circulant matrix) Fourier coefficients.

```python
def conv_solve(self, **kwargs):
    [setattr(self, k, v) for k, v in kwargs.items()]
    # needed for optimization

    if self.fft_type == 0:  # raster with DFS
        E_conv_all, o_E_conv_all = to_conv_mat_raster_discrete(self.
ucell, self.fourier_order[0], self.fourier_order[1], device=self.
device, type_complex=self.type_complex, improve_dft=self.improve_dft)

    elif self.fft_type == 1:  # raster with CFS
        E_conv_all, o_E_conv_all = to_conv_mat_raster_continuous(self
.ucell, self.fourier_order[0], self.fourier_order[1], device=self.
device, type_complex=self.type_complex)

    elif self.fft_type == 2:  # vector with CFS
        E_conv_all, o_E_conv_all = to_conv_mat_vector(self.
ucell_info_list, self.fourier_order[0], self.fourier_order[1],
type_complex=self.type_complex)

    else:
        raise ValueError

    de_ri, de_ti, layer_info_list, T1, kx_vector = self._solve(self.
wavelength, E_conv_all, o_E_conv_all)

    self.layer_info_list = layer_info_list
    self.T1 = T1
    self.kx_vector = kx_vector

    return de_ri, de_ti
```

Code 8: 'conv_solve()'

```python
def solve(self, wavelength, e_conv_all, o_e_conv_all):
    de_ri, de_ti, layer_info_list, T1, kx_vector = self._solve(
        wavelength, e_conv_all, o_e_conv_all)

    # internal info. for the field calculation
    self.layer_info_list = layer_info_list
    self.T1 = T1
    self.kx_vector = kx_vector

    return de_ri, de_ti
```

Code 9: 'solve()'

```python
field_cell = mee.calculate_field(res_x=100, res_y=100, res_z=100)
```

Code 10: 'calculate_field()'

### H.3.2 MAXWELL'S EQUATIONS COMPUTATION

After generating the convolution matrix, meent solves Maxwell's equations and returns diffraction efficiencies with the method 'solve()'. As in the Code 9, it is a wrapper of '_solve()' method that actually does the calculations and returns the diffraction efficiencies with other information that is necessary for the field calculation.

Input parameters:

***wavelength  : float***
 The wavelength of the incident light in vacuum.

***e_conv_all  : array of {float or complex}***
 A stack of convolution matrices of the permittivity array; this is $[\![\varepsilon_{r,g}]\!]$ in Chapter G. The order of the axes is the same as of ucell ($Z\ Y\ X$).

***o_e_conv_all  : array of {float or complex}***
 A stack of convolution matrices of the one-over-permittivity array; this is $[\![\varepsilon_{r,g}^{-1}]\!]$ in Chapter G. The order of the axes is the same as of ucell ($Z\ Y\ X$).

The diffraction efficiencies are 1D array for 1D and 1D-conical grating and 2D for 2D grating.

### H.3.3 FIELD CALCULATION

The 'calculate_field()' method in Code 10 calculates the field distribution inside the structure. Note that the 'solve()' method must be preceded. This function returns 4 dimensional array that the length of the last axis varies depending on the grating type as shown in Code 11. 1D TE and TM has 3 elements (TE has Ey, Hx and Hz in order and TM has Hy, Ex and Ez) while the others have 6 elements (Ex, Ey, Ez, Hx, Hy and Hz) as in Figure 22. Input parameters:

***res_x  : integer***
 The field resolution in X direction (number of split which the period of x is divided by).

***res_y  : integer***
 The field resolution in Y direction (number of split which the period of y is divided by).

***res_z  : integer***
 The field resolution in Z direction (number of split in thickness of each layer).

***field_algo  : integer***
 The level of vectorization for the field calculation. Default is 2 which is fully vectorized for fast calculation while 1 is half-vectorized and 0 is none. Option 0 and 1 are remained for debugging or future development (such as parallelization).

  • 0: Non-vectorized

  • 1: Semi-vectorized: in X and Y direction

  • 2: Vectorized: in X, Y and Z direction

```
1    # 1D TE and TM case
2    field_cell = torch.zeros((res_z * len(layer_info_list), res_y, res_x,
      3), dtype=type_complex)
3
4    # 1D conincal and 2D case
5    field_cell = torch.zeros((res_z * len(layer_info_list), res_y, res_x,
      6), dtype=type_complex)
```

Code 11: the shape of returned array from 'calculate_field()'

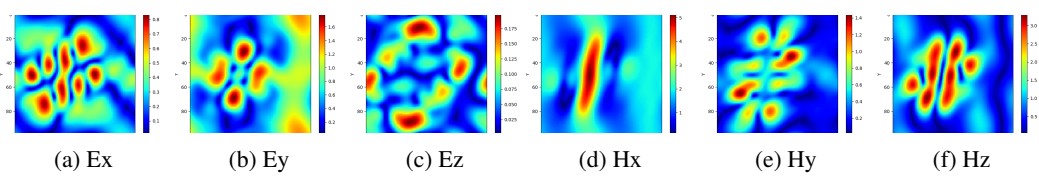

| (a) Ex | (b) Ey | (c) Ez | (d) Hx | (e) Hy | (f) Hz |

Figure 22: **Field distribution on XY plane from 2D grating structure.** (a)-(c): absolute value of the electric field in each direction, (d)-(f): absolute value of the magnetic field in each direction.

# I BENCHMARK

In this section, we will address the 1D metasurface problem covered in the previous work (Seo et al., 2021) with `meent` so that we can benchmark and analyze its capability and functionality.

## I.1 CASE APPLICATION

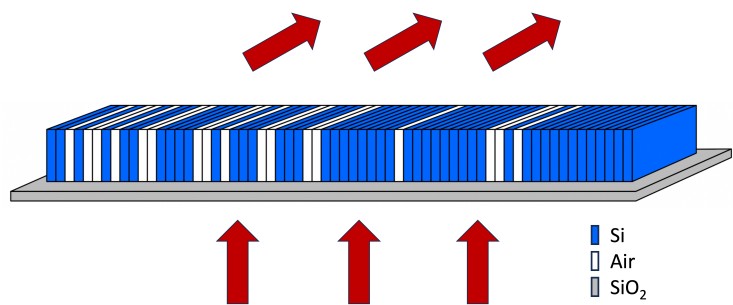

Figure 23: **The image of 1D diffraction metagrating on a silicon dioxide substrate.**

This metagrating deflector is composed of silicon pillars placed on a silica substrate. The device period is divided into 64 cells, and each cell can be filled with either air or silicon. The Figure of Merit for this optimization is set to the deflection efficiency of the $+1^{st}$ order transmitted wave when TM polarized wave is normally incident from the silica substrate as in Figure 23.

## I.2 FOURIER SERIES IMPLEMENTATIONS

When the sampling frequency of permittivity distribution is not enough, Fourier coefficients from DFS is aliased. It can be resolved by increasing the sampling rate that is implemented in the way of duplicating the elements so the array is extended to have identical distribution but larger array size. We will call this Enhanced DFS, and it's implemented in `meent` as a default option.

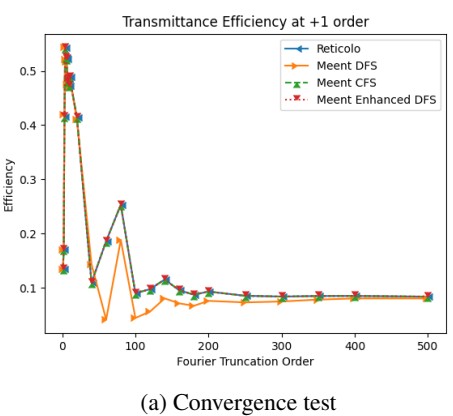
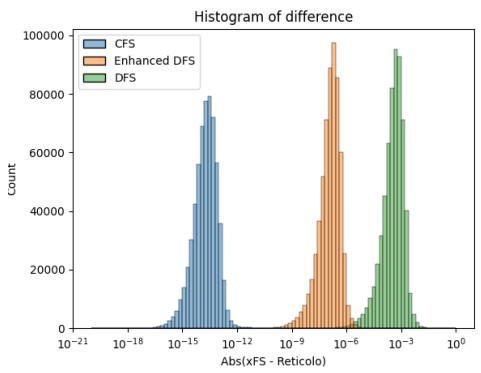

(a) Convergence test  (b) Histogram of the difference compared to Reticolo

Figure 24: **Evaluation of 4 different RCWA implementations: Reticolo, `meent` DFS, `meent` CFS and `meent` Enhanced DFS.** (a) shows diffraction efficiency at $+1^{st}$ order by FTO sweep of a particular structure. DFS behaves differently while the other results seem similar. (b) is the histogram of 600k simulation result (deflection efficiency) difference. Here Reticolo is the reference and other 3 implementations in `meent` are benchmarked.

Figure 24a illustrates the convergence tests of a particular structure with four different RCWA implementations. Considering Reticolo as the reference, we can see CFS is well-matched but DFS shows different behavior. This is due to the insufficient sampling rate of permittivity distribution, which can be resolved by Enhanced DFS. Figure 24b is the histogram of the discrepancies from Reticolo result. About 600k structures were evaluated with 4 implementations and the errors of 3 `meent` implementations were calculated based on Reticolo. CFS

Table 15: Performance test condition

| backend | device | bit | alpha server | beta server | gamma server |
|---------|--------|-----|--------------|-------------|--------------|
| NumPy | CPU | 64 | (A1) | (B1) | (C1) |
| NumPy | CPU | 32 | (A2) | (B2) | (C2) |
| JAX | CPU | 64 | (A3) | (B3) | (C3) |
| JAX | CPU | 32 | (A4) | (B4) | (C4) |
| JAX | GPU | 64 | - | (B5) | (C5) |
| JAX | GPU | 32 | - | (B6) | (C6) |
| PyTorch | CPU | 64 | (A7) | (B7) | (C7) |
| PyTorch | CPU | 32 | (A8) | (B8) | (C8) |
| PyTorch | GPU | 64 | - | (B9) | (C9) |
| PyTorch | GPU | 32 | - | (B10) | (C10) |

Figure 25: **Performance test: calculation time with respect to FTO.** Top row is the result from 64bit and bottom is from 32bit. The first column is the result from the test server alpha and the rest is beta and gamma in order.

shows the smallest errors and this is because Reticolo too uses CFS (CFS algorithms in `meent` are adopted from Reticolo). Enhanced DFS decreases the error about three orders of magnitudes (e.g., the median of DFS is 4.3E-4 and this becomes 1.4E-7).

### I.3 COMPUTING PERFORMANCE

In this section, computing options to speed up the calculation - backend, device (CPU and GPU) and architecture (64bit and 32bit) - will be benchmarked. Table 5 is the hardware specification of the test server and Table 15 is the index of each test condition.

The graphs in Figure 25 are calculation time vs FTO with all the data per machine and architecture. Before look into the details, we will briefly mention some notice in this figure. (1) JAX can't afford large FTO regardless of device. We suspect that this is related to JIT compilation which takes much time and memory for the compilation at the first run. (2) GPU with JAX and PyTorch can't accept large FTO even though GPU memory is more than needed for array upload. (3) if large amount of calculation is needed, Numpy or PyTorch on CPU is the option. (4) no golden option exists: it is recommended to find the best option for the test environment by doing benchmark tests.

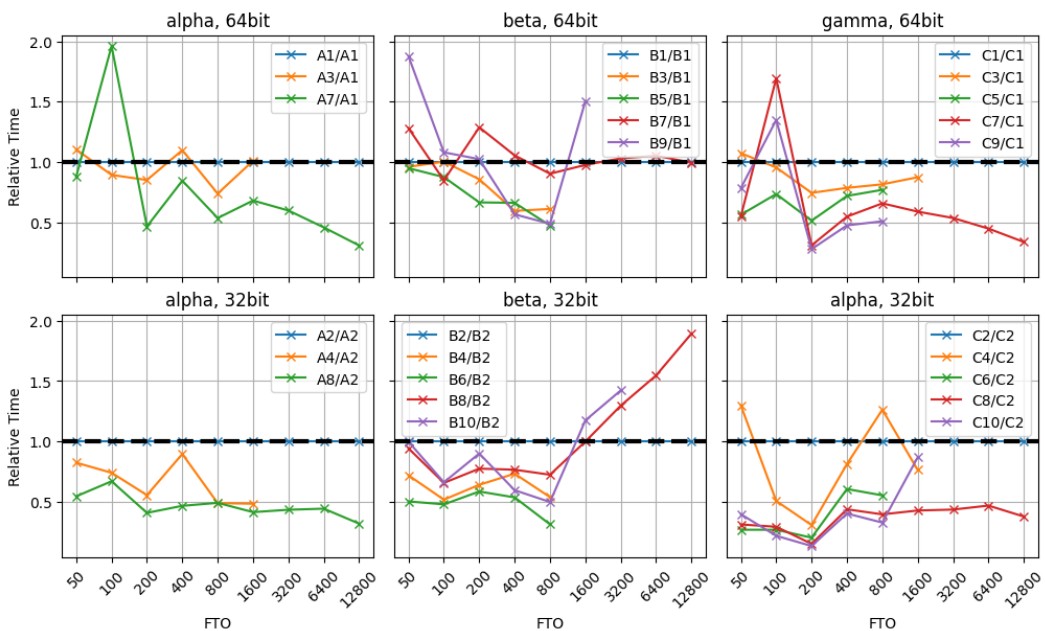

Figure 26: **Performance test: calculation time by FTO sweep.** The result is normalized by NumPy case from the same options to compare the behavior of other backends. In these plots, black dashed line is $y = 1$ and the results of NumPy cases lie on this line since they are normalized by themselves.

We will visit these computing options one by one. The option C9 at FTO 1600 will be excluded in further analyses: this seems an optimization issue in PyTorch or CUDA.

### I.3.1 BACKEND: NUMPY, JAX AND PYTORCH

NumPy, JAX and PyTorch as a backend are benchmarked. NumPy is installed via PyPI which is compiled with OpenBLAS. There are many types of BLAS libraries and the most representative ones are OpenBLAS and MKL (Math Kernel Library). As of now, PyPI provides NumPy with OpenBLAS while conda does one with MKL. This makes small discrepancy in terms of speed and precision hence pay attention when doing consistency test between machines. Figure 26 is the relative simulation time per server and architecture normalized by the time of NumPy case in the same conditions to make comparison easy. In small FTO regime, all the options were successfully operated and no champion exists. Hence it is strongly recommended to run benchmark test on your hardware and pick the most efficient one. In case of X7 (A7, B7 and C7), Alpha and Gamma show the same behavior - spike in 100 - while beta shows fluctuation around B1. One possible reason for this is the type of CPU. The CPUs of Alpha and Gamma belong to 'Xeon Scalable Processors' group but Beta is 'Xeon E Processors'. Currently we don't know if this actually makes difference or some other reason (such as the number of threads or BLAS implementation) does. This result may vary if MKL were used instead of OpenBLAS. In large FTO, only two options are available: NumPy and PyTorch on CPU in 64 bit. In case of JAX, the tests were failed: we watched memory occupation surge during the simulation which seems unrelated to matrix calculation. This might be an issue of JIT (Just In Time) compilation in JAX. Between NumPy and PyTorch, PyTorch is about twice faster than NumPy in both architectures at Alpha and Gamma, but beta shows different behavior. This too, we don't know the root cause but one notable difference is the family of CPU type.

### I.3.2 DEVICE: CPU AND GPU

Figure 27 shows the relative simulation time of GPU cases normalized by CPU cases on the same backend and architecture. Note that it is **relative** time, so the smaller time does not mean it is a good option for the simulation experiments: the relative time can be small even if the absolute time of CPU and GPU are very large compared to other options.

JAX shows good GPU utilization throughout the whole range (except one point in beta) regardless of the architecture. Considering the architecture, the data trend in beta is not clear while the gamma clearly shows that GPU utilization can be more effective in 32bit operation. PyTorch data is a bit noisier than of JAX, but has the similar behavior per server. The data in beta is hard to conclude as the JAX cases and the gamma too shows ambiguous trend but we can consider GPU option is efficient with wide range of FTOs.

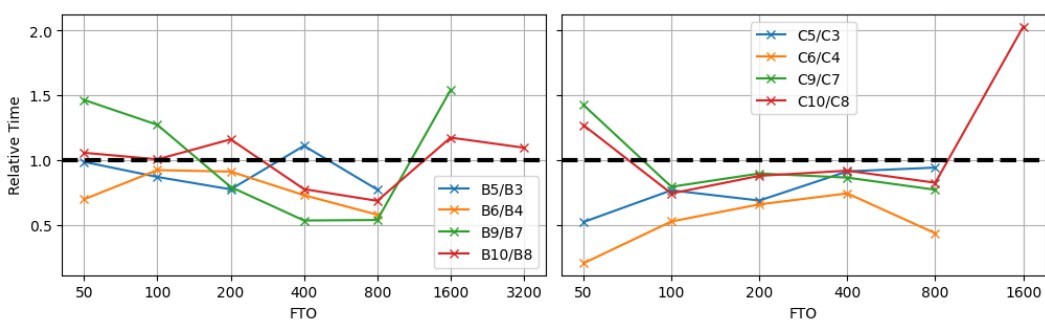

Figure 27: **Performance test result.** The calculation time of GPU cases are normalized by CPU cases from the same options to see the efficiency of GPU utilization. In these plots, black dashed line is $y = 1$ where the capability of both are the same.

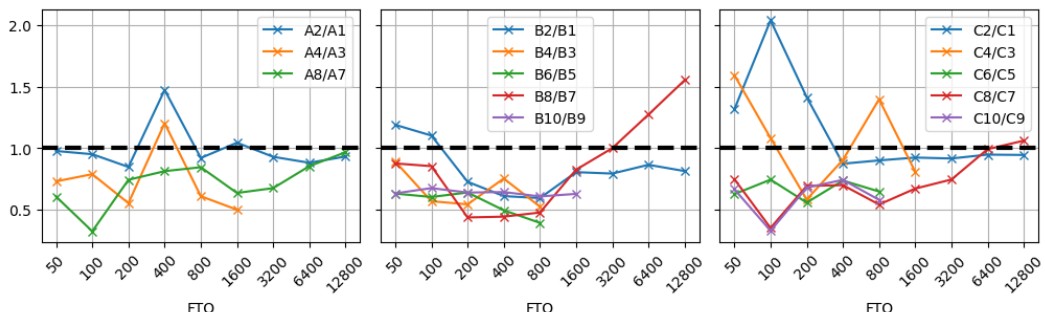

Figure 28: **Performance test result.** The calculation time of 32bit cases are normalized by 64bit cases from the same options. In these plots, black dashed line is $y = 1$ where the capability of both are the same.

Up to date, eigendecomposition for non-hermitian matrix which is the most expensive step ($O(M^3 N^3)$) in RCWA, is not implemented on GPU in JAX and PyTorch hence the calculations are done on CPU and the results are sent back to GPU. As a result, we cannot expect great performance enhancement in using GPUs.

### I.3.3 ARCHITECTURE: 64 AND 32 BIT

In Figure 28, calculation time of 32bit case is normalized by 64bit case with the same condition. With some exceptions, most points show that simulation in 32bit is faster than 64bit. Here are some important notes: (1) From our understanding, the eigendecomposition (Eig) in NumPy operates in 64bit regardless of the input type - even though the input is 32bit data (float32 or complex64), the matrix operations inside Eig are done in 64bit but returns the results in 32bit data type. This is different from JAX and PyTorch - they provides Eig in 32bit as well as 64bit. Hence the 32bit NumPy cases in the figure approach to 1 as FTO increases because the calculation time for Eig is the same and it is the most time-consuming step. (2) Keep in mind that 32bit data type can handle only 8 digits. This means that $1000 + 0.00001$ becomes $1000$ without any warnings or error raises. For such a reason, the accuracy of 32bit cases in the figures are not guaranteed - we only consider the calculation time. (3) Eig in PyTorch shows interesting behavior: as FTO increases, calculation time in 32bit overtakes 64bit - see A8/A7, B8/B7 and C8/C7. This is counter-intuitive and we don't have good explanation but cautiously guess that this might be related to the accuracy and precision in Eig or an optimization issue of PyTorch.

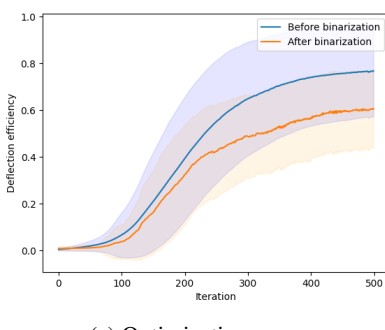
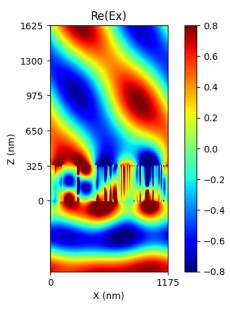

(a) Optimization curve          (b) Electric field

Figure 29: **Optimization result of 1D beam deflector.** (a) The deflection efficiencies are calculated for every iterations and this experiment is repeated 100 times with random starting points (b) Electric field distribution from the final structure

## J  APPLICATIONS

`meent` is expected to be useful for solving inverse design and optimization problems in OCD and metasurface design. In this section, we present some exemplary cases where `meent` proves its capabilities. Leveraging the automatic differentiation function, we successfully carry out optimization for diffraction gratings and achieve inverse design of the geometric parameters.

### J.1  INVERSE DESIGN OF 1D DIFFRACTION GRATING

In this example, we will optimize 1D beam deflector that was used for benchmark in Chapter I using AD with these options - 256 cells, FTO $= 100$, $\lambda_0 = 900$ nm and the deflection angle $= 50°$. During the optimization, each cell can have non-binary refractive index values, leading to a gray-scale optimization. To obtain the final structure consisting of only silicon/air binary structures, an additional binary-push process is required. The initial structure for optimization is randomly generated so the each cell can have the refractive index value between of air and silicon under uniform distribution. The Figure of Merit for this optimization process is set to the $+1^{st}$ order diffraction efficiency, and the gradient is calculated by AD. The refractive indices are updated over multiple epochs using the ADAM optimizer (Kingma & Ba, 2017) with the learning rate of 0.5.

Figure 29a shows the deflection efficiency change by iteration. Two solid lines are averaged value of all the samples at the same iteration step. Shaded area is marked with $\pm$ standard deviation from the average. The blue line (Before binarization) is the result of device with any real number between two refractive indices (silicon and air), which is non-practical, and the orange line (After binarization) is the final device composed of silicon and air. The best result we found is 89.4%.

### J.2  INVERSE DESIGN OF 2D DIFFRACTION GRATING

Here, we demonstrate optimization of a 2D diffraction metagrating as shown in Figure 30a. Similar to the previous 1D diffraction metagrating, the 2D diffraction metagrating also consists of silicon pillars located on top of a silicon dioxide substrate. TM polarized wave with $\lambda = 1000$ nm is normally incident from the bottom of the substrate and the device is designed to deflect the incident light with deflection angle $\theta = 60°$ in $X$-direction. The device has a rectangular unit cell of period $\lambda/\sin\theta \approx 1150$ nm and $\lambda/2 = 500nm$ for the x and y-axis, respectively. Moreover, the unit cell is gridded into $256 \times 128$ cells which is either filled by air or silicon. The convergence of RCWA simulation for different number of Fourier harmonics are plotted in Figure 30b. Considering the trade-off between simulation accuracy and time, we set $N_x = 13$ and $N_y = 10$.

After 110 epochs of optimization, the final structure achieves an efficiency of 92% and successfully deflects the incoming beam at a $60°$ angle (Figure 30d). The optimized structure and the learning curve are presented in Figure 30a and Figure 30c, respectively.

### J.3  INVERSE DESIGN OF 1D GRATING COLOR ROUTER

Until now, we have focused on the problems where the FoM was simply defined. However, in this example, we aim to demonstrate the optimization process of a meta color router, which involves a complex FoM.

A meta color router is an optical component designed for next-generation image sensors. It is designed to route the incoming light to the subpixel region of corresponding color, as depicted in Figure 31. In this exemplary case,

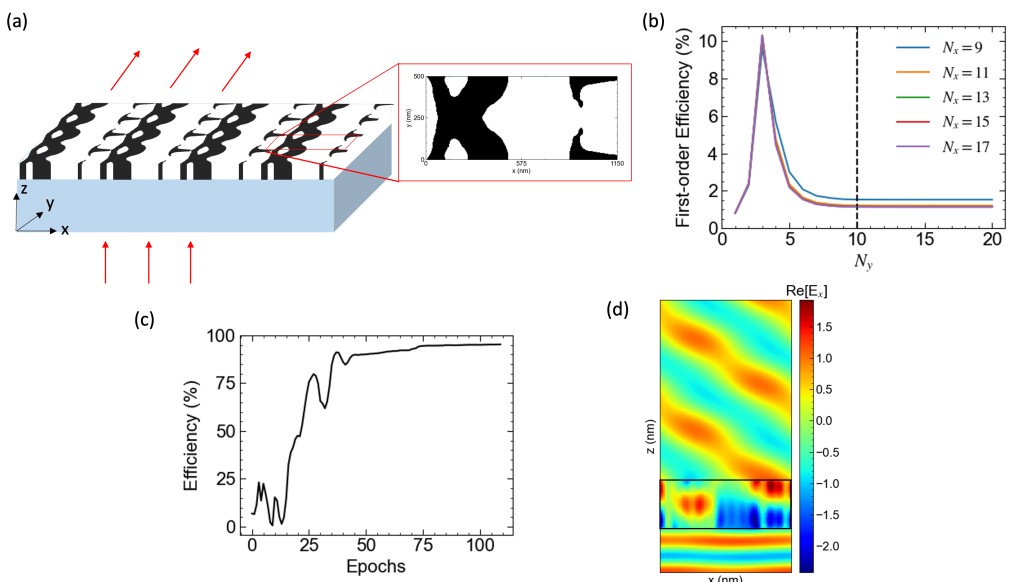

Figure 30: **Optimization result of 2D beam deflector.** (a) The schematic of 2D beam deflector and the final structure after optimization. (b) Convergence test of the initial structure. (c) Learning curve of structure optimization for 110 epochs. Spatial blurring and binary push is applied on each epoch (d) The electric field distribution of the optimized structure in XZ plane.

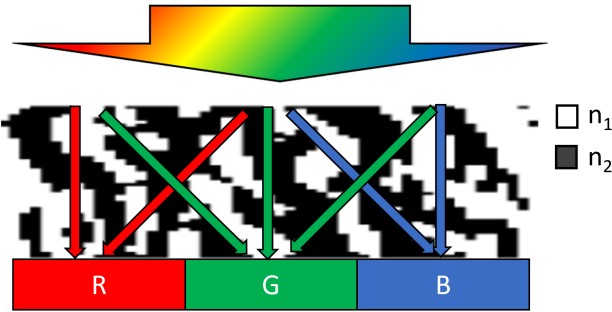

Figure 31: **A schematic of a color router.** The incoming light is guided to subpixels of corresponding wavelength.

we consider an RGB meta color router featuring a pixel pitch of $0.5\mu m$ that consists of vertically stacked 1D binary gratings. The constituent dielectrics are silicon dioxide and silicon nitride, with fixed refractive indices of 1.5 and 2.0, respectively. The meta device region (width of $1.5\mu m$ and height of $2\mu m$) is sliced into 8 layers with 64 cells per layer.

The FoM for this meta color router is defined as the average of TE mode electric field intensity over the corresponding subpixel region, as given by Equation equation 76.

$$FoM = \frac{1}{N} \sum_{\lambda_1}^{\lambda_N} \frac{\int_{x_1}^{x_2} |\vec{\mathbf{E}}(\lambda)|^2 dx}{\int_0^P |\vec{\mathbf{E}}(\lambda)|^2 dx} \times T(\lambda) \tag{76}$$

Here, $\mathbf{E}$ represents the electric field within the subpixel region, while $T$ represents transmittance. The parameter $x \in (x1, x2)$ determines the desired subpixel region corresponding to the incident beam wavelength. For simplicity, we define the wavelength ranges for R, G, and B as 600 nm - 700 nm, 500 nm - 600 nm, and 400 nm - 500 nm, respectively. Throughout the optimization process, optical efficiencies are averaged across 9 wavelength points to ensure a finely tuned broadband response.

The optimization procedure for the meta color router follows a similar approach to the previous examples, including random initialization, optimization via back-propagated gradients with or without binary push. The optimization curve and the final binarized device structure are shown in Figure 32.

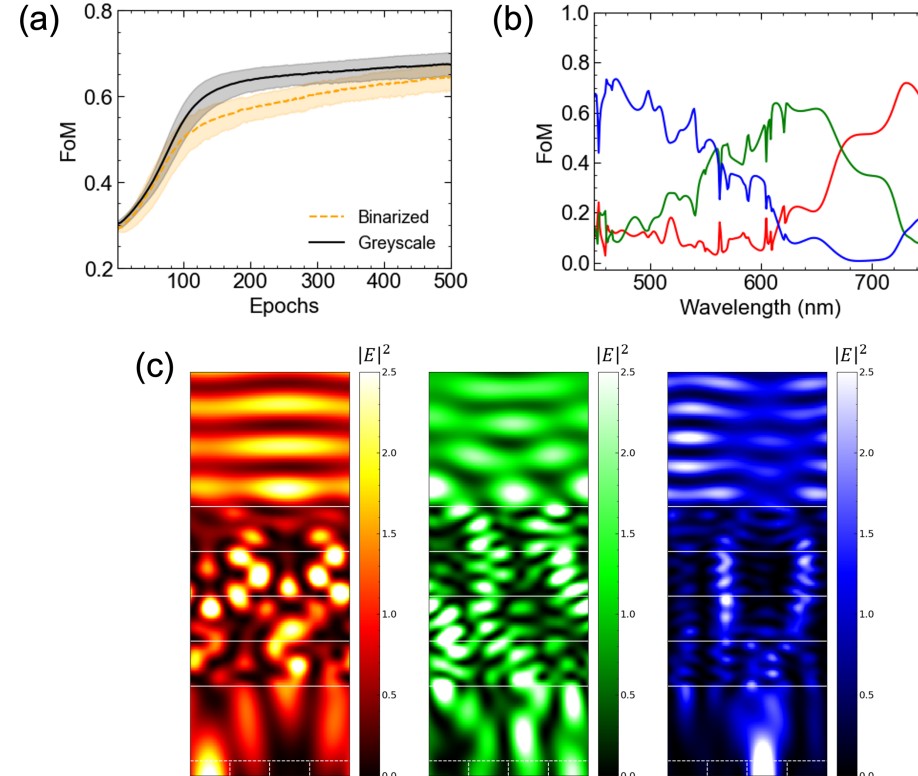

Figure 32: **Optimization result of 1D grating meta color router.** (a) Optimization curve of greyscale device and binary-pushed device at each epoch. (b) Color sorting efficiency spectrum. (c) The electric field inside the final color router device.

# K  LICENSES

**Development**

- Numpy: BSD license
- JAX: Apache License 2.0
- PyTorch: BSD license (BSD-3)

**Experiment**

- Ray: Apache License 2.0
- SheepRL: Apache License 2.0
- neuraloperator: MIT license

