# OpenReview forum: "MEENT: DIFFERENTIABLE ELECTROMAGNETIC SIMULATOR FOR MACHINE LEARNING"
_ICLR.cc/2025/Conference — Submitted to ICLR 2025_

### Official Review · Reviewer_8JQs · 2024-10-26

**Soundness:** 2
**Presentation:** 3
**Contribution:** 1
**Rating:** 3
**Confidence:** 5

**Summary:**

This paper introduces Meent, a software designed for electromagnetic field simulation using RCWA, and highlights potential applications in machine learning (ML) through Meent. The paper is clear and easy to follow. See below for detailed comments.

**Strengths:**

The paper is easy to follow.

**Weaknesses:**

A key contribution of this work is the development of Meent. However, it is important to note that Meent is purely software based on numerical algorithms, and its development does not involve any ML techniques. The only connection to ML is the authors' use of Meent as a simulation backbone for dataset generation or device optimization.

This raises a significant concern: **there is no substantial contribution to the field of ML in this work**. If I may respectfully provide some examples that would align with ICLR's focus on ML, especially in the context of ML for photonics or AI for science: (1) The authors could propose a new neural operator (or architecture) that outperforms existing approaches such as FNO in predicting electromagnetic fields, such as [1,2]. (2) The paper could introduce a novel reinforcement learning algorithm that demonstrates better performance than current RL methods or use RL addressing a problem no one has done before, such as AlphaFold, AlphaGo. (3) The authors could apply ML techniques to design a more compact photonic device with much better performance that no one can.

Unluckily, this work doesn't fall into any of the categories above. While I recognize the potential value of this work, the primary contribution is not centered on ML. Therefore, I must recommend rejection for ICLR and suggest the authors consider alternative venues that might be more suitable, such as IEEE JLT, APL Photonics, Optica, Optica Express, Nature Photonics, or Nanophotonics.

[1] Jiaqi Gu et al., 'Neurolight: A physics-agnostic neural operator enabling parametric photonic device simulation,' Neurips 2022.

[2] Pingchuan Ma et al., 'PIC2O-Sim: A Physics-Inspired Causality-Aware Dynamic Convolutional Neural Operator for Ultra-Fast Photonic Device FDTD Simulation,' Arxiv.

**Questions:**

See 'Weakness' section.

---

> ### Author Response · Authors · 2024-11-20
>
> We thank reviewer 8JQs for taking the time to read our paper and for the helpful comments, along with the kind suggestions of examples.
>
> > A key contribution of this work is the development of Meent. However, it is important to note that Meent is purely software based on numerical algorithms, and its development does not involve any ML techniques. The only connection to ML is the authors' use of Meent as a simulation backbone for dataset generation or device optimization.
> >
>
> Our work combines innovative contributions in both optics and machine learning, each with its own distinct novelty.
>
> **Optics Contribution:**
>
> From an optics perspective, our work introduces the first Python implementation that simultaneously supports vector methods for shape optimization and automatic differentiation. This is a significant advancement because vector methods allow for modeling in continuous space, whereas traditional raster methods operate in discrete space. The ability to optimize in continuous space with automatic differentiation is a powerful tool that can enable more accurate and efficient designs in optical systems.
>
> **Machine Learning Contribution:**
>
> In the field of machine learning, our novel contribution lies in its application to the physics domain. While there have been some studies applying neural operators to nanoscale metagrating structures, to our knowledge, no previous research has explored model-based reinforcement learning in the context of partial differential equations (PDEs) combined with control problems. A particularly exciting outcome of our work is the observation that the world model can explicitly learn the dynamics of electromagnetism, notably the prediction of electric fields in our specific problem setup. This achievement represents a promising step forward in making reinforcement learning models in the PDE plus control domain both explainable and interpretable.
>
> > If I may respectfully provide some examples that would align with ICLR's focus on ML, especially in the context of ML for photonics or AI for science:
> >
>
> "Applications to the physical sciences" can take many forms, and we believe our work fits within this broad category. While some papers focus on demonstrating the value of their methods for specific applications, our goal is to make a broader impact on both the optics and machine learning communities. We aim to present something new, accessible, and versatile, which fosters mutual understanding between the two fields. By doing so, we hope to spark discussions that lead to meaningful future collaborations and advancements.

---

> > ### Comment · Reviewer_8JQs · 2024-11-20
> > **Thank the authors for the response**
> >
> > I appreciate the authors' detailed response; however, I regret to say that I remain unconvinced regarding the "Machine Learning Contribution". My concerns are outlined below:
> >
> > (1) While I acknowledge that using model-based reinforcement learning (RL) for partial differential equations (PDEs) might be novel to the best of my knowledge, model-based RL is not new in addressing control or real-world problems. For instance, in the case of integrated circuits (an ordinary differential equation, ODE), similar approaches have been applied, as exemplified in [1]. The differences between applying RL to PDEs versus ODEs do not appear to be significant enough to constitute a distinct contribution. If the authors intend to emphasize this aspect, I must respectfully express that my impression of the marginality of the ML contribution has been further reinforced. Also, what is the novelty in the RL algorithm? Do the authors use a standard one?
> >
> > [1] "A Hierarchical Adaptive Multi-Task Reinforcement Learning Framework for Multiplier Circuit Design," ICML, 2024.
> >
> > (2) The connection drawn between the concept of the world model and the specific work presented in this manuscript appears tenuous. Therefore, the claim that "the world model can explicitly learn..." seems inaccurate in this context.
> >
> > (3) Even if we accept the use of the world model concept in this work, predicting electromagnetic (EM) fields is not a novel task, as evidenced by the references I previously provided.

---

> ### Author Response · Authors · 2024-11-21
>
> We appreciate the feedback from reviewer 8JQs.
>
> > Also, what is the novelty in the RL algorithm? Do the authors use a standard one?
> >
>
> We use the standard DreamerV3 algorithm [2].
>
>
> > The differences between applying RL to PDEs versus ODEs do not appear to be significant enough to constitute a distinct contribution.
> >
>
> Our work specifically focuses on PDE in optics, where demystifying complexity of applying RL to PDEs versus ODEs is out of our scope.
>
> ODE plus control problem in circuit design is indeed tackled with their novel reinforcement learning approach (HAVE) in [1]. However as the authors of [1] stated in the paper,
>
> > In this paper, our HAVE falls into the model-free category.
> >
>
> HAVE falls into model-free algorithm, which we believe is very different from model-based approach. Model-based RL entails “explicit” dynamics model  $p≈p_θ(s_{t+1}|s_t,a_t)$, as explained Section 4.2 of out paper.
>
> On the other hand, generally, there is no such explicit dynamics model in the context of model-free RL, leading to different formulation of algorithm such as policy gradient algorithm [3]. Figure 7 of our paper was intended to show the capability of $p_{\theta}$ to predict electric field when modified with agent's action. The electric field was then reconstructed and compared with ground truth explicitly. However, as reviewer 8JQs pointed out, in-depth anatomy and analysis of the $p_{\theta}$ may be conducted in the future work.
>
> [1] "A Hierarchical Adaptive Multi-Task Reinforcement Learning Framework for Multiplier Circuit Design," ICML, 2024.
>
> [2] Hafner, Danijar, et al. "Mastering diverse domains through world models." *arXiv preprint arXiv:2301.04104* (2023).
>
> [3] Sutton, Richard S., et al. "Policy gradient methods for reinforcement learning with function approximation." *Advances in neural information processing systems* 12 (1999).

---

### Official Review · Reviewer_1WHZ · 2024-10-29

**Soundness:** 3
**Presentation:** 3
**Contribution:** 2
**Rating:** 5
**Confidence:** 2

**Summary:**

A differentiable electromagnetic simulation framework (called "meent"), which is able to operate on a continuous space. Moreover, authors have presented six different applications for how to use "meent" as a tool to generate data for ML as well as a solver for inverse problems.

**Strengths:**

- Clear presentation
- Comprehensive set of applications, including investigating machine learning (ML) algorithms in optics problems, and on development of nanophotonic devices.

**Weaknesses:**

I have a concern about the contribution of this paper. While having access to a user-friendly and differentiable software for Physics applications (e.g., EM simulator) is important and definitely helps research communities to accelerate their ideas, I am not completely convinced that this conference is a right place and fit for this paper. The main contribution of this paper is to introduce a python-based software, making the use of other developed tools easier for solving Physics applications. There are certainly other great venues where readers can take advantage of reading this paper and might be more fit to the audience.

**Questions:**

Could you either provide a comparison of your method to MaxwellNet (Lim & Psaltis, 2022)  for electric field prediction, or explain why MaxwellNet was not included as a baseline?"

---

> ### Author Response · Authors · 2024-11-20
>
> We would like to thank reviewer 1WHZ for taking the time to read our paper and for providing helpful comments, particularly regarding the introduction of MaxwellNet, which is a very interesting work involving a physics-based model in the optics domain.
>
> > I have a concern about the contribution of this paper. While having access to a user-friendly and differentiable software for Physics applications (e.g., EM simulator) is important and definitely helps research communities to accelerate their ideas, I am not completely convinced that this conference is a right place and fit for this paper. The main contribution of this paper is to introduce a python-based software, making the use of other developed tools easier for solving Physics applications. There are certainly other great venues where readers can take advantage of reading this paper and might be more fit to the audience.
> >
>
> We present a novel application in the physics domain. While a few studies have applied neural operators to nanoscale metagrating structures, to our knowledge, no study has yet explored model-based reinforcement learning in the context of partial differential equations (PDEs) combined with control problems. It is exciting to observe that our experimental results show the world model can explicitly learn the dynamics of electromagnetism, particularly the prediction of the electric field in our problem setup. We believe this work paves the way for more explainable and interpretable reinforcement learning in the PDE-plus-control domain.
>
> "Applications to the physical sciences" can take many forms, and we believe our work fits within this broad category. While some papers focus on demonstrating the value of their methods for specific applications, our goal is to make a broader impact on both the optics and machine learning communities. We aim to present something new, accessible, and versatile, which fosters mutual understanding between the two fields. By doing so, we hope to spark discussions that lead to meaningful future collaborations and advancements.

---

> > ### Comment · Reviewer_1WHZ · 2024-11-25
> >
> > Thanks for providing more explanation. I'll keep my score as before.

---

### Official Review · Reviewer_AduX · 2024-11-02

**Soundness:** 3
**Presentation:** 3
**Contribution:** 4
**Rating:** 6
**Confidence:** 2

**Summary:**

This work presents meent, a framework that aims to integrate EM simulators into the ML pipelines. In particular, meent contains a differentiable, Python-native EM simulator. The authors demonstrate the value of meent through three concrete applications: (1) generating datasets to train the neural operators; (2) enabling RL-based design of nanophotonic device; (3) constructing solutions for  inverse-problems

**Strengths:**

- Combining ML with EM simulator is an important problem.
- The paper is organized really well and well-written. Starting with the technical details of meent, the authors also present concrete applications. This helps significantly in illustrating the value of meent.
- The contribution of the proposed framework has been clearly discussed and the comparison with existing packages is comprehensive.

**Weaknesses:**

- Missing related work: *Benchmarking Data-driven Surrogate Simulators for Artificial Electromagnetic Materials.* Thirty-fifth Conference on Neural Information Processing Systems Datasets and Benchmarks Track (Round 2)
- I think it would be helpful to present a set of experiments comparing the efficiency of meent. In particular, how fast does meent generate EM simulation comparing to existing Cpp-based methods?
- In other words, does the whole ML+EM pipeline take longer time by incorporating differentiable EM simulation? If the extra time is considerable, then the benefits of meent should be evaluated more carefully.
- Since meent has some simulation error as discussed in Section 3, what is the performance degrade comparing to ML + classical EM simulators (e.g., Reticolo)? I think **it is helpful conduct the same set of experiments with ML+ Reticolo** and compare the resultant performance with the ones under meent.

**Questions:**

See weakness.

---

> ### Author Response · Authors · 2024-11-20
>
> We express our gratitude to AduX for the thorough review and in-depth comments. Your invaluable feedback helped us improve our work, and your kind suggestions on related works enriched our paper.
>
> > Missing related work: *Benchmarking Data-driven Surrogate Simulators for Artificial Electromagnetic Materials.* Thirty-fifth Conference on Neural Information Processing Systems Datasets and Benchmarks Track (Round 2)
> >
>
> We sincerely appreciate the kind advice. We will add it in the related works as you suggested.
>
> > I think it would be helpful to present a set of experiments comparing the efficiency of meent. In particular, how fast does meent generate EM simulation comparing to existing Cpp-based methods?
> >
>
> We have performed simulation time benchmarks - comparing to reticolo, which is great code with versatile features written in MATLAB.
>
> While ours supports automatic differentiation within JAX and PyTorch frameworks, the time for calculation itself is similar but reticolo takes additional (constant) time to call MATLAB api in Python process.
>
> Table of throughput ratio (speed of meent / speed of reticolo) by FTO sweep.
>
> | FTO | 10 | 20 | 40 | 60 | 80 | 100 | 120 | 140 | 160 | 180 | 200 |
> | --- | --- | --- | --- | --- | --- | --- | --- | --- | --- | --- | --- |
> | ratio of throughput | 22.32 | 13.28 | 5.78 | 3.16 | 2.5 | 2.06 | 1.72 | 1.55 | 1.5 | 1.39 | 1.22 |
>
> In this table, we can see the throughput is getting similar as the amount of calculation increases (fto increases) which means the portion of import time is smaller.
>
> > In other words, does the whole ML+EM pipeline take longer time by incorporating differentiable EM simulation? If the extra time is considerable, then the benefits of meent should be evaluated more carefully.
> >
>
> We appreciate for the careful advice and suggestion. Adding ML chains in EM simulation did not show meaningful simulation time increase.
>
> Adding ML chain (for backpropagation) helps to get the gradient. In photonics, it is common to use adjoint method to calculate the gradient of figure of merit with respect to the input - the pixels. Adjoint method enables this with 2 EM simulatinos. But this can be further decreased with automatic differentiation to 1 EM simulation and 1 backpropagation.
>
> Moreover, there are some applications that adjoint method cannot be used such as color router where the adjoint source cannot be defined. Here automatic differentiation is the only option to calculate the gradient.
>
> > Since meent has some simulation error as discussed in Section 3, what is the performance degrade comparing to ML + classical EM simulators (e.g., Reticolo)? I think **it is helpful conduct the same set of experiments with ML+ Reticolo** and compare the resultant performance with the ones under meent.
> >
>
> In Section 3, we demonstrated the discrepancy between the simulation results from Meent and Reticolo. By applying the same Fourier analysis algorithm (Continuous Fourier Series, CFS), we observed a median difference on the order of 1E-14. We believe this level of difference is too small to have any meaningful impact on machine learning modeling.

---

> > ### Comment · Reviewer_AduX · 2024-11-26
> >
> > I appreciate the authors’ response which has addressed my concerns.

---

### Official Review · Reviewer_Jrdo · 2024-11-03

**Soundness:** 2
**Presentation:** 3
**Contribution:** 2
**Rating:** 5
**Confidence:** 3

**Summary:**

- This work emphasizes the significance of electromagnetic simulation in the analysis and design of photonic structures.
- The integration of machine learning into electromagnetic simulation has emerged as a promising solution, with the optics research community increasingly leveraging machine learning algorithms.
- The paper introduces meent, a user-friendly electromagnetic simulation software developed in Python that utilizes rigorous coupled-wave analysis and automatic differentiation.
- It includes three key applications of meent.

**Strengths:**

- It is generally well-written.
- It introduces machine learning researcher-friendly electromagnetic simulation software.
- It provides several interesting applications of meent.

**Weaknesses:**

- I don't know what the technical novelty of this work is.

**Questions:**

- What is the technical novelty of this work?
- This work might be not fit to ICLR, because it proposes a simulation tool.  I would like to discuss this issue with the authors, reviewers, and area chairs.

**Details Of Ethics Concerns:**

There are no ethics concerns for this work.

---

> ### Author Response · Authors · 2024-11-20
>
> We would like to thank reviewer Jrdo for taking the time to read our paper and for providing helpful comments.
>
> > What is the technical novelty of this work?
> >
>
> Our work combines innovative contributions in both optics and machine learning, each with its own distinct novelty.
>
> **Optics Contribution:**
>
> From an optics perspective, our work introduces the first Python implementation that simultaneously supports vector methods for shape optimization and automatic differentiation. This is a significant advancement because vector methods allow for modeling in continuous space, whereas traditional raster methods operate in discrete space. The ability to optimize in continuous space with automatic differentiation is a powerful tool that can enable more accurate and efficient designs in optical systems.
>
> **Machine Learning Contribution:**
>
> In the field of machine learning, our novel contribution lies in its application to the physics domain. While there have been some studies applying neural operators to nanoscale metagrating structures, to our knowledge, no previous research has explored model-based reinforcement learning in the context of partial differential equations (PDEs) combined with control problems. A particularly exciting outcome of our work is the observation that the world model can explicitly learn the dynamics of electromagnetism, notably the prediction of electric fields in our specific problem setup. This achievement represents a promising step forward in making reinforcement learning models in the PDE plus control domain both explainable and interpretable.
>
> We believe this work not only demonstrates the potential of combining optics and ML, but also opens up new avenues for future research in both fields, particularly in solving complex, real-world problems with enhanced understanding and control.
>
> > This work might be not fit to ICLR, because it proposes a simulation tool. I would like to discuss this issue with the authors, reviewers, and area chairs.
> >
>
> We present both a novel application and a simulation tool, which we believe fit well within the broader category of 'applications to the physical sciences,' recognizing that these applications can take many forms.
>
> While some papers focus on demonstrating their value in specific applications, our goal is to make a broader impact on both the optics and machine learning communities. We aim to present something new, accessible, and versatile to both fields, fostering mutual understanding and encouraging dialogue. This, we hope, will spark discussions that lead to meaningful collaborations and future advancements.

---

> > ### Comment · Reviewer_Jrdo · 2024-11-20
> >
> > Thank you for your response.
> >
> > Three of four reviewers pointed out the similar weakness on the fitness of this work for ICLR.  I think that this valid issue has not been resolved by the authors' comment.
> >
> > For the optics contribution mentioned by the authors, this is obviously not the scope of ICLR.  As pointed out by Reviewer 8JQs, some optics or photonics journals might be fit more.
> >
> > For the machine learning contribution mentioned, I believe that the current form does not focus on addressing this contribution.  Maybe the authors should overhaul the submission overall in order to emphasize this contribution.
> >
> > I think that if the authors want to contribute to both communities emphasizing two orthogonal directions to optics and machine learning, NeurIPS Datasets and Benchmarks Track or Journal of Data-centric Machine Learning Research (DMLR) might be a potential choice for publication.

---

### Meta-Review · Area_Chair_myAd · 2024-12-25

**Metareview:**

The paper presents MEENT a differentiable electromagnetic simulation software, that can be used to generate synthetic datasets for ML surrogates training or for using the simulator as part of the training in an ML pipeline.

Reviewers unanimously agreed that this is a great scientific contribution but they questioned the scope of this submission to ICLR, and found it would be more  appropriate submission to scientific journals. Given that the machine learning contribution of this paper is rather not highlighted in the current draft I concur with the authors.

**Additional Comments On Reviewer Discussion:**

Authors and reviewers discussed the paper.

---

### Decision · Program_Chairs · 2025-01-22

Reject